# PreDiff: Leveraging Data Priors to Enhance Time Series Generation with Scarce Samples

## Abstract

The fundamental motivation for time series generation tasks lies in addressing the pervasive challenge of data scarcity. However, we have identified a critical limitation: existing time series generation models are prone to substantial performance degradation when trained on limited data. To tackle this issue, we propose a novel framework that integrates data priors to enhance the robustness and generalization of time series generation under data-scarce conditions. Our framework is structured around a two-stage pipeline: pre-training and fine-tuning. In the pre-training stage, the model is trained on synthetic time series datasets to learn data priors, which encode the fundamental statistical properties and temporal dynamics of time series data. Subsequently, during the fine-tuning stage, the model is refined using a small-scale target dataset to adapt to the specific distribution of the target domain. Extensive experimental evaluations demonstrate that our framework mitigates performance degradation caused by data scarcity, achieving state-of-the-art results in time series generation tasks. This work not only advances the field of time series modeling but also provides a scalable solution for real-world applications where data availability is often limited.

## 1 Introduction

Time series generation (TSG) stands as a cornerstone problem across diverse domains, including finance, energy, and healthcare (Lim & Zohren, 2021). Recent advancements (Goodfellow et al., 2014; Kingma & Welling, 2013; Chen et al., 2018) in machine learning have propelled significant progress in this area, with diffusion models emerging as a particularly promising approach due to their capacity to model intricate temporal dependencies and produce high-fidelity sequences (Ho et al., 2020; Galib et al., 2024; Yuan & Qiao, 2024; Coletta et al., 2023; Narasimhan et al., 2024a). However, the efficacy of these models is heavily contingent upon access to large-scale, high-quality datasets—a requirement that is often impractical in real-world settings.

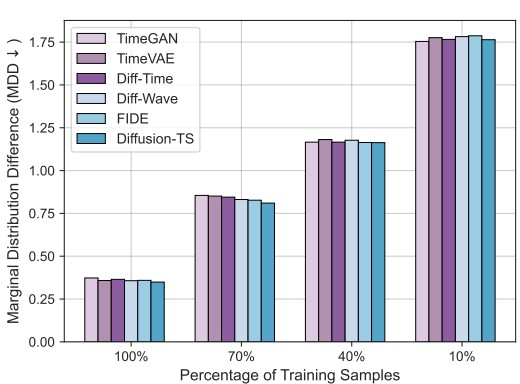

Figure 1: Performance drop in time series generation models on the Stock dataset with varying size.

In numerous practical scenarios, the acquisition of extensive time series data is hindered by privacy constraints (Alaa et al., 2021), prohibitive collection costs, or the infrequency of specific events. Consequently, time series generation under data-scarce conditions has emerged as a critical yet underexplored research frontier (Li et al., 2024a). Traditional models, when applied directly to small-sample datasets, frequently exhibit substantial performance degradation, as shown in Figure 1.

To tackle the challenge of data scarcity, we leverage large-scale time series datasets as data priors. We posit that these prior datasets may encapsulate partial yet informative characteristics of the target datasets. Grounded in this hypothesis, we introduce a novel two-stage training framework built upon diffusion models. During the first stage, diffusion model undergoes pre-training on a synthetic

dataset to learn the prior distribution of time series data, akin to the role of a foundational model. In the second stage, the model is fine-tuned on a limited target dataset, enabling the adaptation of the prior distribution to the target distribution. By decoupling the training process into these two distinct phases, our experimental results demonstrate that the proposed framework achieves superior utilization of synthetic priors and scarce real-world data. The key contributions of this work are summarized as follows:

- We uncover a critical yet frequently overlooked issue: state-of-the-art time series generation models suffer from severe performance degradation under data scarcity, highlighting a significant gap in existing research. Thus, we introduce `PreDiff`, an innovative paradigm for time series generation designed to address data scarcity.

- By integrating data priors with a meticulously crafted pre-training and fine-tuning strategy, `PreDiff` enhances the modeling of target data distributions and effectively mitigates performance degradation induced by limited data availability.

- We find that: 1) A larger data prior improves the performance of `PreDiff`; 2) `PreDiff` effectively handles both trend-dominated and periodic time series, even under extreme data scarcity; 3) `PreDiff` is flexible, integrating various diffusion-based TSG modules and data priors. 4) We also investigate the impact of different data priors. The higher the feature overlap between the data prior and the target data, the better the performance.

## 2 RELATED WORK

Recent advancements in time series generation have explored various architectures, including generative adversarial network (GAN) (Yoon et al., 2019; Seyfi et al., 2022; Wang et al., 2023), variational autoencoders (VAEs) (Desai et al., 2021; Lee et al., 2023; Li et al., 2023; Naiman et al., 2023), Transformer (Li et al., 2024b), and hybrid approaches (Rubanova et al., 2019; Kidger et al., 2021; Zhou et al., 2023; Alaa et al., 2021), often combined with neural networks like long short-term memory (LSTM) (Hochreiter & Schmidhuber, 1997), gated recurrent unit (GRU) (Cho et al., 2014), and Transformer (Vaswani et al., 2017). These methods effectively capture temporal patterns but often struggle with challenges such as mode collapse and instability.

Diffusion models (Ho et al., 2020), originally designed for image generation, have gained traction in time series generation due to their interpretability and ability to model complex data distributions. Methods like Diff-TS (Yuan & Qiao, 2024), DiffWave (Kong et al., 2021), TIME WEAVER (Narasimhan et al., 2024a), and ImagenTime (Naiman et al., 2024) PaD-TS (Li et al., 2025) leverage diffusion processes to enhance temporal pattern modeling. FIDE (Galib et al., 2024) further extends diffusion models to handle rare and extreme events by incorporating frequency-domain strategies. (Jing et al., 2024) synthesizes time series by manipulating existing time series. Additionally, constrained generation approaches, such as those by (Coletta et al., 2023; Narasimhan et al., 2024b), address specific constraints but often overlook the challenge of data scarcity.

Despite these advancements, the issue of time series generation under data scarcity remains understudied. This gap motivates our work, as data scarcity can significantly hinder model performance. We propose `PreDiff` to address this challenge, emphasizing the importance of robust generation methods in data-limited scenarios. Although relevant work (Gonen et al., 2025) is being done to address this issue, it heavily relies on the distribution of the training dataset and suffers from poor generalization (See also Appendix K).

## 3 BACKGROUND

**Problem Definition**    Let $X_{1:\tau} = (x_1, \ldots, x_\tau) \in \mathbb{R}^{\tau \times d}$ denote a time series spanning $\tau$ time steps, where $d$ is the dimensionality of the observed signals. The goal of TSG is to learn a generative model $\mu_\theta$ that maps from a latent space $\mathcal{Z}$ to the time series space $\mathcal{X}$, generating new sequences $x_{\text{gen}}$ that align with the real data distribution $P_{real}(x)$:

$$x_{\text{gen}} = \mu_\theta(z), \quad z \sim P_z, \tag{1}$$

where $P_z$ is a prior distribution over the latent space (e.g., Gaussian).

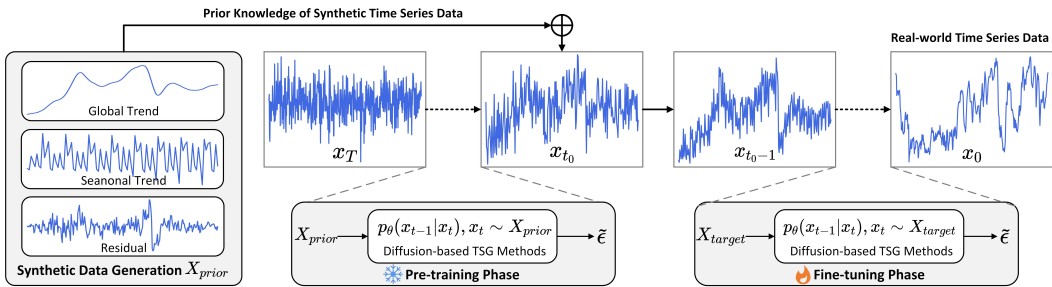

Figure 2: Framework of `PreDiff`, where ❄ means we only update the model parameters from step $T$ to step $t_0$ in pre-training phase. 🔥 means that we train the model parameters from step $t_0$ to step $0$ in the fine-tuning phase, while keeping the parameters frozen in the pre-training phase.

**Challenges in Small-Sample Regimes** When the time series length $\tau$ is small or the dataset is limited, traditional generative methods (e.g., training directly on the entire dataset) face several challenges: 1) **Overfitting Risk**: Models trained on scant data tend to memorize specific sequences rather than learn the true distribution, yielding poor generalization and low-quality outputs. 2) **Insufficient Diversity**: With few examples, generators fail to reproduce the full variability of the real data, producing samples that underrepresent the range of true patterns and incur distributional shifts. 3) **Weak Temporal Modeling**: Complex dependencies—trends, seasonality, or irregular fluctuations—are difficult to capture when data are scarce, leading to synthetic series that mischaracterize the underlying temporal structure.

**Diffusion Process for TSG** Diffusion-based methods for TSG typically involve two processes: the *forward process* and the *reverse process*.

*Forward Process*. A sample $x_0 \sim q(x)$ from the data distribution is gradually corrupted into standard Gaussian noise $x_T \sim \mathcal{N}(0, I)$ through a series of diffusion steps. The transition at each step $t$ is parameterized by: $q(x_t|x_{t-1}) = \mathcal{N}(x_t; \sqrt{1-\beta_t}x_{t-1}, \beta_t I)$, where $\beta_t \in (0, 1)$ controls the amount of noise added at step $t$.

*Reverse Process*. A neural network learns to gradually denoise the sample through the reverse transition: $p_\theta(x_{t-1}|x_t) = \mathcal{N}(x_{t-1}; \mu_\theta(x_t, t), \Sigma_\theta(x_t, t))$, where $\mu_\theta(x_t, t)$ and $\Sigma_\theta(x_t, t)$ are learned parameters. The reverse process is trained to approximate the true posterior $q(x_{t-1}|x_t)$.

**Training Objective** Following (Ho et al., 2020), the denoising model $\mu_\theta(x_t, t)$ is trained using a weighted mean squared error (MSE) loss: $\mathcal{L} = \mathbb{E}_{t,x_0,\epsilon}\left[\|\epsilon - \epsilon_\theta(x_t, t)\|^2\right]$, where $\epsilon_\theta(x_t, t)$ predicts the noise added during the forward process. This objective can be interpreted as optimizing a weighted variational lower bound on the data log-likelihood.

## 4 METHOD: PREDIFF

Figure 2 illustrates the framework of `PreDiff`, which consists of three main components: (1) the construction of a time series prior dataset $X_{prior}$, (2) pre-training of a diffusion model based on $X_{prior}$, and (3) fine-tuning on a small-scale target dataset $X_{target}$.

**Synthesis of $X_{prior}$** Data scarcity often leads to reduced diversity in the target dataset. To mitigate the impact of data scarcity during the pre-training phase, we construct a prior dataset $X_{prior}$ to cover a broader data distribution. The construction of $X_{prior}$ is flexible: it can be derived from large-scale real-world datasets (e.g., Time-MOE (Shi et al., 2024) and Monash dataset (Godahewa et al., 2021)) or synthetic datasets (e.g., ForecastPFN (Dooley et al., 2024)). Specifically, ForecastPFN decomposes synthetic time series data into periodic, global, and noise components, then models these components using mathematical formulations. Unless otherwise specified, this paper uses synthetic datasets generated by ForecastPFN as the prior to validate the effectiveness of `PreDiff`. The details of ForecastPFN can be found in Appendix A. The impact of the prior dataset on generation performance will be discussed in Section I.

**Pre-training Phase** Traditional diffusion models are typically trained on large datasets. However, training diffusion models directly on limited samples often results in significant performance degradation (see Figure 1). To address this, we pre-train the latter half of the diffusion model (i.e., from the intermediate noise state $t$ to the pure noise state $T$) using the prior dataset $X_{prior}$. Specifically, during each epoch of pre-training, we randomly select a diffusion step $t$ within the interval $[t_0, T]$ and forward diffuse $x_t \sim X_{prior}$ to $x_T$. For the forward diffusion method $p$, we default to the approach proposed by (Ho et al., 2020), though this can be adjusted based on specific requirements. The model then predicts the noise at step $t$ using the time series generation network $\mu_\theta(x_t, t)$. During this phase, the model optimizes only over randomly selected steps within $[t_0, T]$, and the loss function is defined as:

$$\mathcal{L}_{\text{pre}} = \mathbb{E}_{t \sim [t_0, T], x_t \sim X_{prior}} \left[ \|\epsilon_\theta(x_t, t) - \epsilon\|^2 \right], \tag{2}$$

where $t_0$ is the segmentation point.

By incorporating prior knowledge, the model expands the target data distribution during pre-training, leveraging the diverse scenarios of synthetic data to provide a broader modeling space and enhance generation capabilities, particularly when the target data distribution is narrow.

**Fine-tuning Phase** During fine-tuning, we train the first half of the diffusion model (i.e., from the target state 0 to the intermediate noise state $t_0$) using the target dataset $X_{target}$. Specifically, in each epoch of fine-tuning, we randomly select a step $t_f$ within the interval $(0, t_0)$ and forward diffuse $x_0 \sim X_{target}$ to $x_{t_f}$ using the same forward diffusion method $p$. The model then predicts the noise at step $t_f$ using the pre-trained network $\mu_\theta$. During this phase, the model optimizes only over randomly selected steps within $(0, t_0)$, and the loss function is defined as:

---

**Algorithm 1** Training Strategy

---

**Require:** Prior dataset $X_{prior}$ (for pretraining), target dataset $X_{target}$ (for fine-tuning), Diffusion model parameters $\theta$, Number of diffusion steps $T$, Step size $\alpha_t$.

1: **Stage 1: Pretrain on Prior Dataset**
2: Initialize model parameters $\theta$ randomly.
3: **for** each sample $x_0$ in $X_{prior}$ **do**
4:     Randomly choose $t$ between $t_0$ to $T$.
5:     Add noise to $x_0$ to obtain $x_t$:
        $x_t = \sqrt{\alpha_t}x_0 + \sqrt{1 - \alpha_t}\epsilon$, where $\epsilon \sim \mathcal{N}(0, I)$.
6:     Update model $\mu_\theta$ by minimizing pretrain loss $\mathcal{L}_{\text{pre}}$.
7: **end for**
8: **Stage 2: Fine-tune on Target Dataset**
9: Load pretrained parameters $\theta$ from Stage 1.
10: **for** each sample $x_0$ in $X_t$ **do**
11:     Randomly choose $t$ between 0 to $t_0$.
12:     Fine-tune $\mu_\theta$ by minimizing fine-tune loss $\mathcal{L}_{\text{ft}}$.
13: **end for**

---

$$\mathcal{L}_{\text{ft}} = \mathbb{E}_{t_f \sim (0, t_0), x_0 \sim X_{target}} \left[ \|\epsilon_\theta(x_{t_f}, t_f) - \epsilon\|^2 \right]. \tag{3}$$

Fine-tuning on the target dataset enhances the model's perception of the target domain distribution, enabling more accurate generation of time series data that aligns with practical requirements. While the pre-training phase learns the distribution from $t$ to $T$, the fine-tuning phase focuses on modeling the distribution from 0 to $t$, further improving generation precision.

**Pretrain, Finetune, and Sampling Details** During pre-training, perturbations are applied to $X_{prior}$ in the forward process to ensure that the denoising diffusion process leverages the temporal prior distribution in $X_{prior}$. This adjustment enhances the effectiveness of our proposed paradigm in capturing and retaining critical information during pre-training. The fine-tuning process is similar to pre-training, except that it uses the dataset $X_{target}$. Our framework is flexible, allowing various diffusion model variants to be directly integrated into `PreDiff`. As shown in Figure 2, we can replace the Diffusion-based TSG Methods in both the pre-training and fine-tuning phases with methods such as DDPM (Ho et al., 2020) and Diff-TS (Yuan & Qiao, 2024). The sampling process follows the same procedure as DDPM, and Appendix C.2 provides the specific implementation details of our method.

## 5 EXPERIMENTS

The following experiments are designed to address four key questions to demonstrate the effectiveness of the proposed method. 1) **RQ1:** Does `PreDiff` outperform the baseline methods? 2) **RQ2:** Is the performance improvement of `PreDiff` attributed to its two-stage architecture? 3) **RQ3:** Can the generated data be evaluated on downstream tasks to demonstrate its effectiveness? 4) **RQ4:** Is it possible to investigate the applicability of data priors?

## 5.1 EXPERIMENTAL SETUP

**Datasets**. We conduct extensive evaluations on four widely-used public datasets: Stock, Energy, ETTh, and fMRI (Ang et al., 2023). Comprehensive descriptions of these datasets, including their domains, characteristics, and preprocessing steps, are provided in Appendix B. To emulate real-world scenarios where only partial observations are available, we employ a percentage-based splitting strategy (e.g., 10% of the Stock dataset). Specifically, a random split point is selected within the full dataset, and the data is segmented proportionally to reflect the challenges of limited data availability in practical settings.

**Baselines**. To rigorously assess the performance of our proposed method, we compare it against a suite of state-of-the-art time series generation models, including TimeGAN (Yoon et al., 2019), TimeVAE (Desai et al., 2021), DiffWave (Kong et al., 2021), DiffTime (Coletta et al., 2023), Diff-TS (Yuan & Qiao, 2024), and FIDE (Galib et al., 2024). These baselines represent a diverse range of methodologies, from generative adversarial networks and variational autoencoders to advanced diffusion-based approaches. Unless otherwise specified, the size of $X_{prior}$ is set to $100K$. Detailed configurations, hyperparameters, and implementation specifics for each baseline are meticulously documented in Appendix C.

**Evaluation Metrics**. In extreme data-scarce settings, generated time series often lack sufficient quality for reliable forecasting or classification. To overcome this, we employ the TSGBench evaluation framework (Ang et al., 2023), which quantifies both global and local similarities between real and synthetic series (see Appendix D for details). We measure: 1) *Global Similarity Metrics*: MDD (Mean Distribution Distance), ACD (Auto-Correlation Distance), SD (Spectral Distance), and KD (Kernel Distance). 2) *Local Similarity Metrics*: ED (Euclidean Distance) and DTW (Dynamic Time Warping). Smaller values of these metrics are preferred.

Table 1: The comprehensive comparison of `PreDiff` against state-of-the-art time series generation models on the Stock dataset at varying data availability levels (100%, 70%, 40%, and 10%). Red text denotes the best results, and blue text denotes the second-best.

| | Models
Metric | TimeGAN
(2019) | TimeVAE
(2021) | DiffTime
(2023) | DiffWave
(2021) | FIDE
(2024) | Diff-TS
(2024) | PreDiff
(Ours) |
|---|---|---|---|---|---|---|---|---|
| 100% | MDD↓ | 0.373 | 0.358 | 0.365 | 0.357 | 0.359 | 0.349 | 0.356 |
| | ACD↓ | 0.054 | 0.058 | 0.059 | 0.046 | 0.066 | 0.053 | 0.051 |
| | SD↓ | 0.359 | 0.335 | 0.336 | 0.341 | 0.345 | 0.322 | 0.328 |
| | KD↓ | 1.696 | 1.698 | 1.609 | 1.597 | 1.603 | 1.594 | 1.598 |
| | ED↓ | 1.093 | 1.116 | 1.104 | 1.111 | 1.105 | 1.087 | 1.061 |
| | DTW↓ | 2.899 | 2.910 | 2.901 | 2.896 | 2.901 | 2.897 | 2.906 |
| 70% | MDD↓ | 0.855 | 0.851 | 0.845 | 0.831 | 0.827 | 0.810 | 0.797 |
| | ACD↓ | 0.082 | 0.085 | 0.079 | 0.100 | 0.091 | 0.085 | 0.088 |
| | SD↓ | 0.403 | 0.402 | 0.405 | 0.387 | 0.395 | 0.386 | 0.377 |
| | KD↓ | 1.856 | 1.853 | 1.854 | 1.837 | 1.845 | 1.823 | 1.805 |
| | ED↓ | 1.145 | 1.151 | 1.160 | 1.156 | 1.187 | 1.145 | 1.139 |
| | DTW↓ | 2.978 | 2.983 | 2.976 | 2.964 | 2.958 | 2.943 | 2.921 |
| 40% | MDD↓ | 1.166 | 1.181 | 1.166 | 1.177 | 1.164 | 1.163 | 1.158 |
| | ACD↓ | 0.424 | 0.444 | 0.424 | 0.417 | 0.421 | 0.419 | 0.414 |
| | SD↓ | 1.217 | 1.213 | 1.218 | 1.206 | 1.215 | 1.206 | 1.209 |
| | KD↓ | 1.054 | 1.041 | 1.043 | 1.050 | 1.049 | 1.041 | 1.033 |
| | ED↓ | 1.393 | 1.375 | 1.379 | 1.371 | 1.386 | 1.364 | 1.362 |
| | DTW↓ | 3.498 | 3.502 | 3.505 | 3.497 | 3.499 | 3.489 | 3.474 |
| 10% | MDD↓ | 1.754 | 1.776 | 1.766 | 1.782 | 1.787 | 1.764 | 1.724 |
| | ACD↓ | 0.742 | 0.761 | 0.777 | 0.764 | 0.765 | 0.751 | 0.728 |
| | SD↓ | 1.471 | 1.477 | 1.471 | 1.470 | 1.476 | 1.455 | 1.342 |
| | KD↓ | 1.661 | 1.675 | 1.669 | 1.662 | 1.669 | 1.643 | 1.629 |
| | ED↓ | 1.514 | 1.560 | 1.506 | 1.525 | 1.532 | 1.511 | 1.499 |
| | DTW↓ | 3.930 | 3.957 | 3.913 | 3.924 | 3.921 | 3.905 | 3.898 |

Table 2: Comprehensive comparison of `PreDiff` against six baselines across four datasets 10% data availability. Lower values indicate better performance.

| | Models
Metric | TimeGAN
(2019) | TimeVAE
(2021) | DiffTime
(2023) | DiffWave
(2021) | FIDE
(2024) | Diff-TS
(2024) | PreDiff
(Ours) |
|---|---|---|---|---|---|---|---|---|
| Stock | MDD↓ | 1.754 | 1.776 | 1.766 | 1.782 | 1.787 | 1.764 | 1.724 |
| | ACD↓ | 0.742 | 0.761 | 0.777 | 0.764 | 0.765 | 0.751 | 0.728 |
| | SD↓ | 1.471 | 1.477 | 1.471 | 1.470 | 1.476 | 1.455 | 1.342 |
| | KD↓ | 1.661 | 1.675 | 1.669 | 1.662 | 1.669 | 1.643 | 1.629 |
| | ED↓ | 1.514 | 1.560 | 1.506 | 1.525 | 1.532 | 1.511 | 1.499 |
| | DTW↓ | 3.930 | 3.957 | 3.913 | 3.924 | 3.921 | 3.905 | 3.898 |
| fMRI | MDD↓ | 0.558 | 0.552 | 0.564 | 0.563 | 0.555 | 0.547 | 0.524 |
| | ACD↓ | 0.635 | 0.626 | 0.627 | 0.645 | 0.639 | 0.625 | 0.608 |
| | SD↓ | 0.755 | 0.755 | 0.744 | 0.759 | 0.760 | 0.738 | 0.729 |
| | KD↓ | 0.431 | 0.440 | 0.419 | 0.409 | 0.421 | 0.421 | 0.410 |
| | ED↓ | 1.632 | 1.634 | 1.652 | 1.642 | 1.642 | 1.635 | 1.609 |
| | DTW↓ | 7.248 | 7.239 | 7.245 | 7.249 | 7.252 | 7.238 | 7.222 |
| ETTh | MDD↓ | 0.885 | 0.889 | 0.902 | 0.885 | 0.905 | 0.889 | 0.872 |
| | ACD↓ | 1.036 | 1.033 | 1.038 | 1.019 | 1.024 | 1.027 | 1.009 |
| | SD↓ | 0.372 | 0.362 | 0.358 | 0.38 | 0.353 | 0.362 | 0.369 |
| | KD↓ | 1.674 | 1.668 | 1.677 | 1.668 | 1.673 | 1.671 | 1.661 |
| | ED↓ | 1.098 | 1.073 | 1.091 | 1.074 | 1.089 | 1.091 | 1.081 |
| | DTW↓ | 3.066 | 3.057 | 3.070 | 3.065 | 3.074 | 3.064 | 3.049 |
| Energy | MDD↓ | 1.470 | 1.462 | 1.461 | 1.469 | 1.454 | 1.456 | 1.427 |
| | ACD↓ | 0.367 | 0.365 | 0.365 | 0.381 | 0.361 | 0.352 | 0.317 |
| | SD↓ | 0.671 | 0.669 | 0.669 | 0.675 | 0.649 | 0.655 | 0.652 |
| | KD↓ | 2.110 | 2.105 | 2.099 | 2.105 | 2.127 | 2.106 | 2.094 |
| | ED↓ | 1.741 | 1.742 | 1.730 | 1.739 | 1.743 | 1.738 | 1.722 |
| | DTW↓ | 7.230 | 7.235 | 7.232 | 7.217 | 7.234 | 7.228 | 7.216 |

## 5.2 RESULTS (RQ1)

**Study on Stock Dataset** This example experiment on the Stock dataset reveals several intriguing phenomena that provide valuable insights and motivate further investigation through additional experiments. In Table 1, across all data availability levels, `PreDiff` consistently outperforms or matches the performance of other models. `PreDiff` demonstrates strong performance even

under extreme data scarcity (10% data). As data availability decreases, `PreDiff`'s performance degradation is more gradual compared to other models, highlighting its robustness. `PreDiff`'s performance is not limited to a single metric but extends across all six metrics, indicating its ability to capture both global and local properties of time series.

`PreDiff` outperforms all baselines, including recent models like FiDE and Diff-TS, which are specifically designed for time series generation. This underscores the effectiveness of `PreDiff`'s two-stage training framework, which leverages synthetic data for pre-training and fine-tunes on target data.

**`PreDiff` vs. Baselines on 10% Data**  We evaluated `PreDiff` against baseline methods on four real-world datasets with 10% data availability. As shown in Table 2, `PreDiff` consistently outperforms other methods, demonstrating its ability to generate high-quality time series data across diverse domains, even under limited data conditions.

The results unequivocally demonstrate that `PreDiff` sets a new benchmark for time series generation, achieving superior performance across diverse datasets and metrics. Compared with Diff-TS, `PreDiff`'s success highlights the importance of its two-stage training strategy, which combines synthetic data pre-training with target data fine-tuning to enhance generalization and robustness. `PreDiff`'s consistent performance across datasets and metrics makes it a highly practical solution for real-world applications, where data diversity and quality are often limited.

**`PreDiff` vs. Diff-TS**  We further evaluated the performance of Diff-TS and `PreDiff` on four datasets at varying data availability levels (100%, 70%, 40%, and 10%). Here, we employ Diff-TS as the diffusion-based TSG module within `PreDiff` to further validate the effectiveness of the proposed training framework.

The Stock dataset exhibits strong trend characteristics, resulting in significant distributional shifts when split into smaller subsets. In contrast, the ETTh dataset, with its pronounced periodicity, shows minimal distributional changes when divided. As shown in Table 3, the performance of both models degrades more gradually on Stock as data availability decreases. For ETTh, performance remains stable at 100%, 70%, and 40% data levels but drops significantly at 10%.

Notably, `PreDiff` consistently outperforms or matches the performance of Diff-TS across all scenarios. `PreDiff` demonstrates strong performance even under extreme data scarcity (10% data). As data availability decreases, PreDiff's performance degradation is more gradual compared to Diff-TS, highlighting its robustness. `PreDiff` outperforms Diff-TS in nearly all scenarios, particularly under low data availability (10% and 40%). This underscores the effectiveness of `PreDiff`'s two-stage training framework, which leverages synthetic data for pre-training and fine-tunes on target data. These results validate the effectiveness of our proposed framework in handling both trend-dominated and periodic time series data, even under extreme data scarcity.

Additionally, we explore the flexibility of `PreDiff` by integrating different diffusion models as the TSG module. Results (see Appendix F) confirm the adaptability of our approach across different diffusion-based generative models.

Table 3: Comprehensive comparison of `PreDiff` and Diff-TS across four datasets under varying data availability levels.

| Per(%) | | 100% | | 70% | | 40% | | 10% |
|---|---|---|---|---|---|---|---|---|
| Models | | Diff-TS | PreDiff | Diff-TS | PreDiff | Diff-TS | PreDiff | Diff-TS | PreDiff |
| Stock | MDD↓ | **0.349** | 0.356 | 0.810 | **0.797** | 1.163 | **1.158** | 1.764 | **1.724** |
| | ACD↓ | 0.053 | **0.051** | **0.085** | 0.088 | 0.419 | **0.414** | 0.751 | **0.728** |
| | SD↓ | **0.322** | 0.328 | 0.386 | **0.377** | **1.206** | 1.209 | 1.455 | **1.342** |
| | KD↓ | **1.594** | 1.598 | 1.823 | **1.805** | 1.041 | **1.033** | 1.643 | **1.629** |
| | ED↓ | 1.087 | **1.061** | 1.145 | **1.139** | 1.364 | **1.362** | 1.511 | **1.499** |
| | DTW↓ | **2.897** | 2.906 | 2.943 | **2.921** | 3.489 | **3.474** | 3.905 | **3.898** |
| fMRI | MDD↓ | 0.294 | **0.288** | 0.311 | **0.305** | 0.392 | **0.374** | 0.547 | **0.524** |
| | ACD↓ | 0.198 | **0.194** | 0.239 | **0.202** | 0.467 | **0.432** | 0.625 | **0.608** |
| | SD↓ | 0.154 | **0.137** | 0.277 | **0.271** | 0.429 | **0.401** | 0.738 | **0.729** |
| | KD↓ | 0.118 | **0.106** | **0.153** | 0.159 | 0.282 | **0.258** | 0.421 | **0.410** |
| | ED↓ | **0.873** | 0.877 | 0.938 | **0.932** | 1.236 | **1.202** | 1.635 | **1.609** |
| | DTW↓ | 6.088 | **6.071** | 6.216 | **6.201** | 6.623 | **6.594** | 7.238 | **7.222** |
| ETTh | MDD↓ | 0.307 | **0.292** | 0.508 | **0.493** | 0.529 | **0.511** | 0.889 | **0.872** |
| | ACD↓ | 0.201 | **0.188** | 0.227 | **0.209** | 0.295 | **0.286** | 1.027 | **1.009** |
| | SD↓ | 0.217 | **0.206** | 0.225 | **0.213** | 0.257 | **0.239** | **0.362** | 0.369 |
| | KD↓ | 0.598 | **0.574** | **0.664** | 0.673 | 0.739 | **0.721** | 1.671 | **1.661** |
| | ED↓ | **0.821** | 0.823 | 0.902 | **0.870** | 0.917 | **0.908** | 1.091 | **1.081** |
| | DTW↓ | 2.288 | **2.269** | 2.503 | **2.475** | 2.554 | **2.549** | 3.064 | **3.049** |
| Energy | MDD↓ | 0.960 | **0.951** | 1.068 | **1.066** | 1.139 | **1.131** | 1.456 | **1.427** |
| | ACD↓ | 0.243 | **0.237** | **0.287** | 0.289 | 0.327 | **0.319** | 0.352 | **0.317** |
| | SD↓ | 0.324 | **0.308** | 0.378 | **0.372** | 0.463 | **0.446** | 0.655 | **0.652** |
| | KD↓ | 1.431 | **1.418** | 1.756 | **1.739** | 1.969 | **1.930** | 2.106 | **2.094** |
| | ED↓ | **1.046** | 1.048 | 1.153 | **1.151** | 1.346 | **1.329** | 1.738 | **1.722** |
| | DTW↓ | **6.541** | 6.549 | 6.825 | **6.821** | 6.952 | **6.949** | 7.228 | **7.216** |

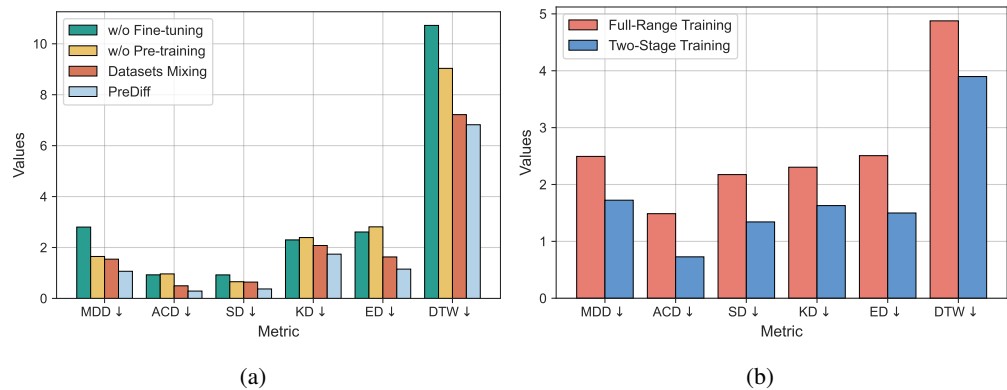

(a)                                   (b)

Figure 3: (a) The performance of Pre-training Only, Fine-tuning Only and `PreDiff` on 70% of the Energy dataset. (b) The performance of full-range training and two-stage training on 10% of the Stock dataset.

## 5.3 ABLATION STUDY (RQ2)

**Validation of Two-stage Strategy**  To validate the effectiveness of the proposed two-stage training framework, we conduct ablation experiments to assess the contribution of each stage by comparing three configurations: (1) **Pre-training Only (w/o Fine-tuning):** Training solely on $X_{prior}$ without target data fine-tuning. (2) **Fine-tuning Only (w/o Pre-training):** Training directly on the target data, omitting pre-training. (3) **Datasets Mixing:** Combine synthetic and target samples into a single training set to test whether performance gains arise merely from increased data volume. (4) **Full Model (`PreDiff`):** Pre-train on prior data, then fine-tune on the target data.

As shown in Figure 3(a), the full two-stage approach consistently outperforms all alternatives across every metric, demonstrating its superior ability to integrate synthetic and real data for high-quality time-series generation under data scarcity.

**Validation of** $t$  To evaluate the impact of the boundary step $t$ in the two-stage training framework on time series generation, we designed the following ablation experiments: (1) **Full-Range Training**: Train the model on $X_{prior}$ to learn the generation process from $0$ to $T$, then continue training on the target data from $0$ to $T$. (2) **Two-Stage Training**: Train the model on $X_{prior}$ to learn the generation process from $t$ to $T$, then fine-tune on the target data from $0$ to $t$.

Experiments conducted on 70% of the Energy dataset (see Figure 3 (b) show that full-range training performs poorly due to its susceptibility to overfitting, which negatively impacts the second training phase. In contrast, the two-stage training framework demonstrates significant advantages by explicitly separating long-term and short-term dynamics. This approach improves the precision and dynamic alignment of generated data while enhancing training efficiency and model generalization.

## 5.4 VALIDATION ON DOWNSTREAM CLASSIFICATION TASK (RQ3)

We assess the utility of our generated time series for downstream classification by using the EEG Eye State dataset (Dua et al., 2017) to create a blink-detection task. Employing a "train on real, test on synthetic" protocol for computing the DS metric, we first train a two-layer LSTM solely on the real data, then evaluate its accuracy on synthetic series produced by various generation methods. At just 10% real-data availability (Table 19), our approach yields statistically significant accuracy gains over all baselines, demonstrating its superior performance in data-scarce downstream scenarios.

Table 4: Classification accuracy in EEG Eye dataset.

| Models | TimeGAN | TimeVAE | DiffTime | DiffWave | FIDE | Diff-TS | PreDiff |
|---|---|---|---|---|---|---|---|
| Accuracy | 0.529 | 0.543 | 0.601 | 0.587 | 0.591 | 0.605 | 0.622 |

## 5.5 APPLICABILITY OF DATA PRIORS (RQ4)

**Effect of Different** $X_{prior}$    To investigate the impact of different prior datasets $X_{prior}$ on the performance of `PreDiff`, we employ ForecastPFN (Dooley et al., 2024), SNIP (Meidani et al., 2023), and the real-world Monash dataset (Godahewa et al., 2021) as priors. The dataset sizes are set to $100K$ and $10M$, respectively. The results are shown in Table 20.

Table 5: The results of `PreDiff` on different $X_{prior}$ datasets with $100K$ and $10M$ data.

| Size | Metric | ForecastPFN | SNIP | Monash |
|------|--------|-------------|-------|--------|
| $100K$ | MDD | 1.724 | 2.016 | **1.718** |
|  | ACD | **0.728** | 0.754 | 0.832 |
|  | SD | **1.342** | 1.359 | 1.351 |
|  | KD | **1.629** | 1.763 | 1.922 |
|  | ED | **1.499** | 1.636 | 1.526 |
|  | DTW | 3.898 | 3.973 | **3.872** |
| $10M$ | MDD | **1.697** | 2.001 | 1.736 |
|  | ACD | **0.736** | 0.772 | 0.851 |
|  | SD | 1.354 | **1.327** | 1.377 |
|  | KD | **1.624** | 1.729 | 1.939 |
|  | ED | **1.485** | 1.685 | 1.511 |
|  | DTW | **3.825** | 3.985 | 3.872 |

Across both dataset sizes, ForecastPFN consistently achieves the best performance in the majority of metrics. We attempt to investigate how different data distributions affect model performance by analyzing visualization results that reflect representation coverage. We quantify the range of representations covered by the datasets generated by ForecastPFN and Monash using a set of statistical metrics, including stationarity (ADF test), forecastability, frequency-domain characteristics (FFT mean), seasonality, trend (Mann–Kendall test), and permutation entropy (see Appendix E for full descriptions). We then use Radviz to visualize the high-dimensional statistical features of 256-length time-series segments drawn from both our synthetic data and the Monash datasets. From Monash—which includes weather, traffic, electricity, tourism, medical, and energy domains—we sample 100K segments per domain. The Radviz visualization (see Figure ??) shows that the distributional diversity of our synthetic dataset is higher than that of the Monash datasets.

When $X_{prior}$ is sampling from ForecastPFN, increasing the dataset size from $100K$ to $10M$ generally improves performance across all priors, as evidenced by lower metric values. Larger datasets consistently improve performance, underscoring the importance of data scale in training effective priors for `PreDiff`.

However, when $X_{prior}$ is drawn from SNIP or Monash, enlarging the prior set actually degrades performance.    This stems from stock data's strong seasonality and trend patterns, which ForecastPFN can faithfully reproduce—thus boosting generation quality. Consequently, one should prioritize priors whose distribution closely matches that of the target data.

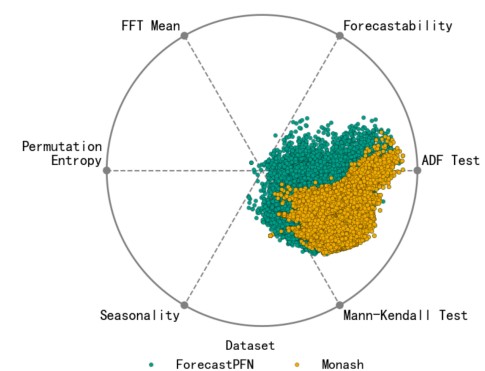

Figure 4: Radviz visualization of the dataset generated from ForecastPFN and Monash dataset.

**Guide for Choice of** $X_{prior}$. The core idea of PreDiff is to extract shared features from data priors to enhance the representations for target tasks in low-resource scenarios. When selecting data priors, we aim to choose those that are highly relevant to the data distribution of the target task. In cases where such priors are unavailable or poorly aligned, we generate synthetic data with more diverse features tailored to the target task—where "features" refer to those most critical to task performance. For example, in time series forecasting, the most influential features are typically seasonality and trend. The data priors used in this work are specifically designed to incorporate diverse seasonal and trend patterns, thereby facilitating the learning of transferable shared features for forecasting tasks.

**Size of** $X_{prior}$. we further investigate the impact of $X_{prior}$ size on the performance of `PreDiff` (See Appendix H). A size of $100K$ may be sufficient for many scenarios, while larger sizes (e.g., $10M$) can be used when higher precision is critical.

**Analysis of** $t_0$. The boundary step $t_0$ in `PreDiff` marks where pre-training on synthetic priors ends and fine-tuning on target data begins. Appendix Table 14 shows that the best $t_0/T$ decreases as the target dataset size grows, reflecting a more diffusion steps are considered in pre-training when richer, larger real datasets better capture the true distribution.

Table 6: Sensitivity of Generation Quality to the Pre-training–Fine-tuning Cut-off Step $t_0/T$ on the Stocks dataset.

| $t_0/T$ | MDD | ACD | SD | KD | ED | DTW |
|---|---|---|---|---|---|---|
| 0.1 | 1.4953 | 4.6021 | 0.5013 | 2.2559 | 1.3189 | 3.3668 |
| 0.2 | 1.4979 | 3.6204 | 0.5376 | 2.3769 | 1.3674 | 3.5219 |
| 0.3 | 1.5995 | 4.6938 | 0.7198 | 2.6055 | 1.3692 | 3.4928 |
| 0.4 | 1.4463 | 3.1264 | 0.7584 | 2.7387 | 1.2130 | 3.1588 |
| 0.5 | 1.5612 | 3.6939 | 0.5974 | 2.2831 | 1.2801 | 3.3264 |
| 0.6 | 1.3992 | 3.1089 | 0.3948 | 2.4328 | 1.3934 | 3.6810 |
| 0.7 | 1.7664 | 4.0166 | 0.6497 | 2.2884 | 1.4189 | 3.6948 |
| 0.8 | 1.5603 | 3.4891 | 0.7104 | 2.2442 | 1.3548 | 3.5688 |
| 0.9 | 1.9687 | 4.8139 | 0.7991 | 2.4227 | 1.1439 | 2.9169 |
| 0.999 | 1.8266 | 4.6018 | 0.6569 | 2.3816 | 1.1549 | 2.8831 |

## 5.6 COMPREHENSIVE COMPARISON AGAINST IMAGENFEW (GONEN ET AL., 2025)

To comprehensively evaluate the robustness of the proposed framework under varying data scarcity, we conduct experiments on three benchmark datasets: Energy, Stocks, and ETTh. We compare the performance of ImagenFew with the baseline `PreDiff` across four different data availability ratios (100%, 70%, 40%, and 10%). Unlike `PreDiff`, which requires additional pre-training and synthetic data generation, ImagenFew directly adapts to limited real data without relying on auxiliary supervision.

Table 7: Comprehensive comparison of ImagenFew against `PreDiff` across three datasets under varying data availability levels (100%, 70%, 40%, and 10%). Best results are highlighted in red.

| Dataset | Metrics | 100% | | 70% | | 40% | | 10% | |
|---|---|---|---|---|---|---|---|---|---|
| | | ImagenFew | PreDiff | ImagenFew | PreDiff | ImagenFew | PreDiff | ImagenFew | PreDiff |
| Energy | MDD↓ | **0.314** | 0.951 | **0.339** | 1.066 | **0.399** | 1.131 | **0.487** | 1.427 |
| | ACD↓ | 0.444 | **0.237** | 0.738 | **0.289** | 1.281 | **0.319** | 2.256 | **0.317** |
| | SD↓ | 0.425 | **0.308** | 0.527 | **0.372** | 0.526 | **0.446** | 1.504 | **0.652** |
| | KD↓ | 12.953 | **1.418** | 21.889 | **1.739** | 17.901 | **1.930** | 151.784 | **2.094** |
| | ED↓ | 1.094 | **1.048** | **1.146** | 1.151 | **1.253** | 1.329 | **1.462** | 1.722 |
| | DTW↓ | 6.610 | **6.549** | 6.872 | **6.821** | 7.346 | **6.949** | 8.447 | **7.216** |
| Stocks | MDD↓ | **0.288** | 0.356 | **0.318** | 0.797 | **0.365** | 1.158 | **0.401** | 1.724 |
| | ACD↓ | 0.116 | **0.051** | 0.114 | **0.088** | 0.251 | 0.414 | 0.352 | 0.728 |
| | SD↓ | **0.310** | 0.328 | **0.372** | 0.377 | **0.447** | 1.209 | **0.488** | 1.342 |
| | KD↓ | 2.020 | **1.598** | 2.236 | **1.805** | 2.601 | **1.033** | 2.681 | **1.629** |
| | ED↓ | 1.189 | **1.061** | 1.179 | **1.139** | **1.212** | 1.362 | **1.174** | 1.499 |
| | DTW↓ | 3.060 | **2.906** | 3.031 | **2.921** | **3.126** | 3.474 | **3.055** | 3.898 |
| ETTh | MDD↓ | **0.023** | 0.292 | **0.024** | 0.493 | **0.027** | 0.511 | **0.031** | 0.872 |
| | ACD↓ | 0.244 | **0.188** | 0.287 | **0.209** | 0.358 | **0.286** | **0.487** | 1.009 |
| | SD↓ | **0.085** | 0.206 | **0.026** | 0.213 | **0.079** | 0.239 | **0.164** | 0.369 |
| | KD↓ | **0.364** | 0.574 | **0.149** | 0.673 | **0.383** | 0.721 | **0.186** | 1.661 |
| | ED↓ | 6.026 | **0.823** | 6.094 | **0.870** | 6.078 | **0.908** | 6.148 | **1.081** |
| | DTW↓ | 16.747 | **2.269** | 16.970 | **2.475** | 16.915 | **2.549** | 17.058 | **3.049** |

As shown in Table 21, `PreDiff` demonstrates clear advantages in key metrics such as ACD, ED, and DTW, regardless of the data availability level. These advantages become more pronounced under low-data scenarios, indicating that `PreDiff` is able to capture a relatively complete distribution of real data even in extreme data-scarcity conditions. This, to some extent, reflects the effectiveness of our approach in the pre-training stage, where diverse synthetic data scenarios are leveraged to expand the target data distribution.

Moreover, the performance of `PreDiff` remains relatively stable as the amount of available data decreases, exhibiting stronger robustness. In particular, under the 10% data availability setting, the ACD and ED metrics remain stable (e.g., ACD on the Energy dataset increases only slightly from 0.237 to 0.317), which further validates the effectiveness of its prior knowledge integration mechanism.

To enhance the fairness of the comprehensive comparison experiments, we further adopt the `PreDiff` data partitioning and preprocessing strategy uniformly for both models. From the results

in Appendix Table 10, the performance of ImagenFew exhibits a significant decline after uniformly adopting the data partitioning and preprocessing strategy of `PreDiff`.

It is worth noting that this marked performance drop of ImagenFew confirms its dependence on specific data preprocessing methods. Furthermore, analysis reveals that, compared with the data partitioning and preprocessing strategy of ImagenFew, the design of `PreDiff` in this aspect is more reasonable and realistic: first, in the time series domain, sensor failures and malfunctions often lead to data collection containing only short continuous segments, which is a common scenario of data scarcity. Second, normalizing on the complete dataset before resampling implicitly introduces global statistical features, causing a distribution shift between randomly sampled data and real-world scarce scenarios. This makes it difficult to accurately simulate data scarcity conditions and, to some extent, undermines fairness.

## 6    CONCLUSIONS

In this work, we address the critical challenge of data scarcity in time series generation, a pervasive issue that significantly hampers the performance of generative models. We propose `PreDiff`, a novel two-stage framework that leverages data priors to enhance the robustness and generalization of time series generation under data-limited conditions. By decoupling the training process into pre-training on synthetic datasets and fine-tuning on small-scale target datasets, `PreDiff` effectively bridges the gap between synthetic priors and real-world data distributions.

However, current pretraining uses synthetic data, which may not fully capture complex real-world temporal patterns, particularly in specialized domains (e.g., clinical or financial data). Fine-tuning mitigates gaps between synthetic and real data, but performance may drop if the target distribution diverges significantly from the prior. Although effective for classification, broader validation on tasks like anomaly detection and forecasting would further demonstrate generality.

Future research directions include exploring hybrid priors that combine the strengths of synthetic and real-world datasets, as well as extending `PreDiff` to other domains where data scarcity is a significant challenge. We believe that our work provides a foundational framework for addressing data scarcity in time series generation and inspires further advancements in this important area.

## ETHICS STATEMENT

PreDiff enables high-quality time series generation in data-scarce scenarios, but it may pose societal and ethical risks. Synthetic data could be misused for fraud or to evade detection systems, particularly in financial, medical, or industrial contexts. Moreover, biases in synthetic priors may propagate through the model, leading to unfair performance in downstream tasks. The indistinguishability between real and synthetic data may also challenge data accountability and transparency. Finally, if fine-tuning on real data is not properly controlled, there is a risk of privacy leakage. We recommend responsible dataset design, fairness evaluation, and usage safeguards to mitigate these risks.

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

## A  FORECASTPFN (DOOLEY ET AL., 2024)

Most existing time-series forecasting methods rely on large amounts of training data. However, in real-world applications, initial observations are often very limited, sometimes with only 40 data points or fewer. In this context, the applicability of traditional methods is restricted, while the performance of existing zero-shot forecasting methods heavily depends on the quality and diversity of the pretraining data. This study introduces ForecastPFN, the first zero-shot time-series forecasting model trained entirely on synthetic data distributions. Extensive experiments demonstrate that ForecastPFN achieves superior performance in zero-shot forecasting tasks compared to state-of-the-art methods, with significantly faster inference speeds.

ForecastPFN is a Prior-Data Fitted Network (PFN) trained offline to approximate Bayesian inference. The model requires no training on new data and makes predictions in a single forward pass. This method introduces a new synthetic data generation method. It is a modular synthetic data generation framework was designed to simulate the diversity of real-world time-series data, incorporating multi-scale seasonal trends, global linear and exponential trends, and Weibull-based noise.The synthetic data generation process balances complexity and trainability to ensure robust forecasting performance.

The synthetic time series is modeled as a combination of two independent components: the underlying trend ($\psi$) and noise ($z_t$).Trend: Includes linear and exponential trends to capture global patterns. Seasonality: Captures multi-scale periodic trends (e.g., weekly, monthly, yearly). Noise: Based on the Weibull distribution, designed with an expected value of 1 to ensure independence from trends and seasonality.

The time series $y_t$ is defined as the product of the following components:

$$y_t = \psi(t) \cdot z_t = \text{trend}(t) \cdot \text{seasonal}(t) \cdot z_t$$

- **Trend Component:**

$$\text{trend}(t) = (1 + m_{\text{lin}} \cdot t + c_{\text{lin}}) \cdot \left( m_{\text{exp}} \cdot c_{\text{exp}}^t \right)$$

  where the parameters of the linear and exponential trends (e.g., $m_{\text{lin}}, c_{\text{lin}}, m_{\text{exp}}, c_{\text{exp}}$) are sampled from normal distributions.

- **Seasonal Component:**

$$\text{seasonal}_\nu(t) = 1 + m_\nu \sum_{f=1}^{\lfloor p_\nu/2 \rfloor} \left[ c_{f,\nu} \sin\left(\frac{2\pi f t}{p_\nu}\right) + d_{f,\nu} \cos\left(\frac{2\pi f t}{p_\nu}\right) \right]$$

  where $\nu \in \{\text{week}, \text{month}, \text{year}\}$ represents the time scale, $p_\nu$ is the period length, and $c_{f,\nu}$, $d_{f,\nu}$ are harmonic coefficients.

- **Noise Component:**

$$z_t = 1 + m_{\text{noise}} \cdot (z - \bar{z}), \quad z \sim \text{Weibull}(1, k)$$

  where $m_{\text{noise}}$ and $k$ control the intensity and shape of the noise distribution, respectively.

**Diversity:** The sampling ranges of the parameters are designed to cover a wide variety of possible patterns, simulating diverse real-world time-series behaviors. **Trainability:** The structure of the synthetic data ensures that the model does not diverge or stall during training, despite the diversity in the data. **Balancing Signal and Noise:** Multiplicative noise is used instead of additive noise to maintain a consistent signal-to-noise ratio across different time-series trends.

## B  DATASET

We evaluate our proposed method on four publicly available datasets to demonstrate its effectiveness across diverse domains:

- **Stock**: This dataset comprises daily historical Google stock data from 2004 to 2019, including trading volume, high, low, opening, closing, and adjusted closing prices. It is widely used for financial time series analysis.

- **Energy**: This dataset records the energy consumption of appliances in a low-energy building, providing insights into energy usage patterns.
- **ETTh (Electricity Transformer Temperature)**: A benchmark dataset for long-term time series forecasting (Zhou et al., 2021), containing hourly records of power load, transformer temperature, and other electricity-related metrics.
- **fMRI**: A synthetic dataset for causal discovery, consisting of simulated blood oxygen level-dependent (BOLD) time series. We select a subset with 50 features for our experiments.

Table 8: Dataset Details.

| Dataset | # of Samples | dim | $l$ | Link |
|---------|--------------|-----|-----|------|
| Stocks | 3773 | 6 | 24 | https://finance.yahoo.com/quote/GOOG |
| Energy | 19711 | 28 | 24 | https://archive.ics.uci.edu/ml/datasets |
| ETTh | 17420 | 7 | 24 | https://github.com/zhouhaoyi/ETDataset |
| fMRI | 10000 | 50 | 24 | https://www.fmrib.ox.ac.uk/datasets |

Table 8 shows the statistics of the datasets and all datasets are available online via the link. $l$ is the length of generated time series. To ensure fairness, all datasets are preprocessed using the **TSGBench** pipeline (Ang et al., 2023), which includes data splitting, normalization, and standardization. For experiments involving partial data (e.g., 10% of the Stock dataset), we randomly select a split point and segment the data proportionally to simulate real-world scenarios with limited data availability.

## C  BASELINES AND IMPLEMENTATION DETAIL

### C.1  BASELINES

We compare our method against state-of-the-art models from three categories:

- **GAN-based Models**: TimeGAN, which uses a three-layer GRU architecture and the recommended loss function settings (Yoon et al., 2019).
- **VAE-based Models**: TimeVAE, with a latent dimension of 8 and hidden layer sizes of 50, 100, and 200 (Desai et al., 2021).
- **Diffusion-based Models**: DiffWave (Kong et al., 2021), DiffTime (Coletta et al., 2023), Diff-TS (Yuan & Qiao, 2024), and FIDE (Galib et al., 2024). For these models, we adjust their architectures to ensure comparable parameter counts.

Table 9: Model hyperparameters for different datasets.

| Parameter | Stocks | fMRI | ETTh | Energy |
|-----------|--------|------|------|--------|
| attention heads | 4 | 4 | 4 | 4 |
| attention head dimension | 16 | 16 | 24 | 24 |
| encoder layers | 2 | 4 | 3 | 4 |
| decoder layers | 2 | 4 | 2 | 3 |
| batch size | 64 | 64 | 128 | 64 |
| sample size | 256 | 256 | 256 | 256 |
| timesteps / sampling steps | 1000 | 2000 | 1000 | 2000 |
| training steps | 1000 | 1500 | 1800 | 2500 |

### C.2  IMPLEMENTATION DETAILS

All experiments are conducted on a high-performance machine equipped with an Intel® Core® i9 12900K CPU @ 5.20 GHz, 64 GB RAM, and an NVIDIA GeForce RTX 3090 GPU. This setup ensures reproducibility and fairness across all comparisons.

To evaluate the performance of the proposed method, we compared it with the following baseline models: (1) GAN-based: TimeGAN, (2) VAE-based: TimeVAE, and (3) Diffusion-based: DiffWave, DiffTime, Diffusion-TS, and FIDE. For TimeGAN, we used the loss function settings recommended in the original paper and employed a three-layer GRU architecture. For TimeVAE, we set the latent dimension to 8 and experimented with hidden layer sizes of 50, 100, and 200. For DiffWave, DiffTime, and FIDE, we adjusted the architecture parameters to ensure similar model sizes.

## D    EVALUATION METRICS

We adopt the rigorous evaluation framework of TSGBench, utilizing a comprehensive set of metrics to assess both global and local properties of the generated time series:

### D.1    FEATURE-BASED MEASURES

1. **MDD (Mean Distribution Distance)**: Measures the discrepancy in statistical distributions between generated and real data.
2. **ACD (Auto-Correlation Distance)**: Quantifies the similarity in temporal dependencies.
3. **SD (Spectral Distance)**: Evaluates the alignment in frequency domain characteristics.
4. **KD (Kernel Distance)**: Assesses the similarity in high-dimensional feature spaces.

Strong performance on these metrics indicates that the model effectively captures the underlying data distribution, generating time series that preserve the global statistical and structural properties of the real data.

### D.2    DISTANCE-BASED MEASURES

1. **ED (Euclidean Distance)**: Measures point-wise reconstruction accuracy.
2. **DTW (Dynamic Time Warping)**: Evaluates temporal alignment and shape similarity, accommodating non-linear time shifts.

Superior performance on these metrics demonstrates the model's ability to accurately reconstruct fine-grained temporal patterns, ensuring that the generated sequences align closely with the real data in both timing and morphology.

## E    DATASET STATISTICAL CHARACTERIZATION COVERAGE EXPERIMENTS

**Metric.**    To further examine the diversity of the artificially synthesized data in the $S^2$ dataset, we conduct a sampling assessment from six dimensions: stationarity, predictability, frequency domain characteristics, complexity, seasonality intensity, and trend characteristics. For each dimension, we select corresponding statistical indicators for dataset evaluation and quantification, as detailed below:

1. **Augmented Dickey-Fuller (ADF) Test:** We employ the ADF test to assess the stationarity of time series, using its test statistic as an indicator of time series stationarity Elliott et al. (1992).
2. **Forecastability:** Based on Goerg (2013) method, we determine whether a time series is chaotic or can be accurately predicted through machine learning models by using Fourier decomposition and entropy. Note that since the method provided by Goerg (2013) is only applicable to multivariate time series, we invert the sampled single-channel time series to form a dual-channel series to calculate the indicator.
3. **FFT Mean:** We utilize the average of the Fourier transform power spectrum to evaluate the frequency domain characteristics of time series. This indicator can be used to measure the overall intensity of time series and assess the energy distribution.
4. **Permutation Entropy:** This indicator assesses the dynamic complexity of a time series by analyzing its permutation patterns Cao et al. (2004). We set the embedding dimension $m = 3$ and time delay $\tau = 1$, and calculate its specific value using Shannon Entropy in Equation 4. See Cao et al. (2004) for more detailed calculation.

5. **Seasonality:** We decompose the time series into trend, seasonal and residual components using the Seasonal-Trend Decomposition using LOESS (STL) algorithm Cleveland et al. (1990). Then, we calculate the intensity of the seasonal component in the time series according to Equation 5.

6. **Mann-Kendall Test:** This is a non-parametric statistical method used to detect monotonic trends in time series Esterby (1996). The basic principle is to compare the size relationship between each data point and other data points in the time series. Therefore, this method does not rely on a specific distribution of data and is not affected by outliers. We use the statistical test results of this method as the evaluation indicator, where -1 indicates a downward trend, 1 indicates an upward trend, and 0 indicates no obvious trend.

$$\text{Permutation} = -\sum_{j=1}^{K} P_j \times \ln P_j, \tag{4}$$

$$\begin{cases} Y_t = T_t + S_t + R_t \\ \text{Seasonality} = \max\left\{0, 1 - \frac{\text{Var}(R_t)}{\text{Var}(S_t + R_t)}\right\} \end{cases}, \tag{5}$$

where, $P_i$ represents the frequency of the $i$-th permutation model in the permutation entropy, and $K = m!$ is the total number of permutation patterns Cao et al. (2004). $Y_t$ represents the original time series, $T_t$, $S_t$ and $R_t$ are the trend, seasonal and residual components decomposed by the STL algorithm Cleveland et al. (1990) respectively. $\text{Var}(\cdot)$ means calculating the variance of a series.

## F    FLEXIBILITY OF PREDIFF

Table 10: The Performance of different variants on multiple datasets. Best results are highlighted in red.

| Dataset | Method | DDPM | DiffTime | Diff-TS |
|---------|--------|------|----------|---------|
| Stocks | MDD | 1.747 | 1.756 | 1.724 |
|  | ACD | 0.766 | 0.746 | 0.728 |
|  | SD | 1.438 | 1.402 | 1.342 |
|  | KD | 1.664 | 1.731 | 1.629 |
|  | ED | 1.539 | 1.527 | 1.499 |
|  | DTW | 3.928 | 4.202 | 3.898 |
| fMRI | MDD | 0.588 | 0.612 | 0.524 |
|  | ACD | 0.673 | 0.651 | 0.608 |
|  | SD | 0.801 | 0.779 | 0.729 |
|  | KD | 0.466 | 0.498 | 0.410 |
|  | ED | 1.712 | 1.689 | 1.609 |
|  | DTW | 7.354 | 7.291 | 7.222 |
| ETTh | MDD | 0.894 | 0.911 | 0.872 |
|  | ACD | 1.062 | 1.037 | 1.009 |
|  | SD | 0.402 | 0.385 | 0.369 |
|  | KD | 1.682 | 1.703 | 1.661 |
|  | ED | 1.126 | 1.103 | 1.081 |
|  | DTW | 3.121 | 3.087 | 3.049 |
| Energy | MDD | 1.462 | 1.503 | 1.427 |
|  | ACD | 0.361 | 0.339 | 0.317 |
|  | SD | 0.701 | 0.684 | 0.652 |
|  | KD | 2.128 | 2.153 | 2.094 |
|  | ED | 1.798 | 1.763 | 1.722 |
|  | DTW | 7.284 | 7.259 | 7.216 |

To further validate the flexibility and adaptability of `PreDiff`, we integrate different diffusion models as the TSG module to explore the scalability of the framework across various diffusion-based architectures. Specifically, we conduct experiments under the 10% data availability setting on four datasets—Stocks, fMRI, ETTh, and Energy, and extend the baseline models to include DDPM, Diff-TS and DiffTime.

In implementation, we adopt DDPM with a U-Net architecture and adjust the trainable parameters of both DDPM and DiffTime to match the parameter scale of `PreDiff`, ensuring a fair comparison. Experimental results demonstrate that `PreDiff` can effectively transfer across different diffusion-based generative models while maintaining stable performance advantages, further confirming its generality and applicability to diverse time series generation tasks.

## G    COMPREHENSIVE COMPARISON AGAINST IMAGENFEW

To comprehensively evaluate the robustness of the proposed framework under varying data scarcity, we conduct experiments on three benchmark datasets: Energy, Stocks, and ETTh. We compare the performance of ImagenFew with the baseline `PreDiff` across four different data availability ratios (100%, 70%, 40%, and 10%). Unlike `PreDiff`, which requires additional pre-training and synthetic data generation, ImagenFew directly adapts to limited real data without relying on auxiliary supervision. This enables ImagenFew to maintain stable and competitive performance across both high- and low-resource settings, highlighting its effectiveness in addressing data scarcity.

Table 11: Comprehensive comparison of ImagenFew against `PreDiff` across three datasets under varying data availability levels (100%, 70%, 40%, and 10%). Best results are highlighted in red.

| Dataset | Metrics | 100% | | 70% | | 40% | | 10% | |
|---|---|---|---|---|---|---|---|---|---|
| | | ImagenFew | PreDiff | ImagenFew | PreDiff | ImagenFew | PreDiff | ImagenFew | PreDiff |
| Energy | MDD↓ | **0.314** | 0.951 | **0.339** | 1.066 | **0.399** | 1.131 | **0.487** | 1.427 |
| | ACD↓ | 0.444 | **0.237** | 0.738 | **0.289** | 1.281 | **0.319** | 2.256 | **0.317** |
| | SD↓ | 0.425 | **0.308** | 0.527 | **0.372** | 0.526 | **0.446** | 1.504 | **0.652** |
| | KD↓ | 12.953 | **1.418** | 21.889 | **1.739** | 17.901 | **1.930** | 151.784 | **2.094** |
| | ED↓ | 1.094 | **1.048** | **1.146** | 1.151 | **1.253** | 1.329 | **1.462** | 1.722 |
| | DTW↓ | 6.610 | **6.549** | 6.872 | **6.821** | 7.346 | **6.949** | 8.447 | **7.216** |
| Stocks | MDD↓ | **0.288** | 0.356 | **0.318** | 0.797 | **0.365** | 1.158 | **0.401** | 1.724 |
| | ACD↓ | 0.116 | **0.051** | 0.114 | **0.088** | **0.251** | 0.414 | **0.352** | 0.728 |
| | SD↓ | **0.310** | 0.328 | **0.372** | 0.377 | **0.447** | 1.209 | **0.488** | 1.342 |
| | KD↓ | 2.020 | **1.598** | 2.236 | **1.805** | 2.601 | **1.033** | 2.681 | **1.629** |
| | ED↓ | 1.189 | **1.061** | 1.179 | **1.139** | **1.212** | 1.362 | **1.174** | 1.499 |
| | DTW↓ | 3.060 | **2.906** | 3.031 | **2.921** | **3.126** | 3.474 | **3.055** | 3.898 |
| ETTh | MDD↓ | **0.023** | 0.292 | **0.024** | 0.493 | **0.027** | 0.511 | **0.031** | 0.872 |
| | ACD↓ | 0.244 | **0.188** | 0.287 | **0.209** | 0.358 | **0.286** | 0.487 | 1.009 |
| | SD↓ | **0.085** | 0.206 | **0.026** | 0.213 | **0.079** | 0.239 | **0.164** | 0.369 |
| | KD↓ | **0.364** | 0.574 | **0.149** | 0.673 | **0.383** | 0.721 | **0.186** | 1.661 |
| | ED↓ | 6.026 | **0.823** | 6.094 | **0.870** | 6.078 | **0.908** | 6.148 | **1.081** |
| | DTW↓ | 16.747 | **2.269** | 16.970 | **2.475** | 16.915 | **2.549** | 17.058 | **3.049** |

As shown in Table 9, `PreDiff` demonstrates clear advantages in key metrics such as ACD, ED, and DTW, regardless of the data availability level. These advantages become more pronounced under low-data scenarios, indicating that `PreDiff` is able to capture a relatively complete distribution of real data even in extreme data-scarcity conditions. This, to some extent, reflects the effectiveness of our approach in the pre-training stage, where diverse synthetic data scenarios are leveraged to expand the target data distribution.

Moreover, the performance of `PreDiff` remains relatively stable as the amount of available data decreases, exhibiting stronger robustness. In particular, under the 10% data availability setting, the ACD and ED metrics remain stable (e.g., ACD on the Energy dataset increases only slightly from 0.237 to 0.317), which further validates the effectiveness of its prior knowledge integration mechanism.

To enhance the fairness of the comprehensive comparison experiments, we further adopt the `PreDiff` data partitioning and preprocessing strategy uniformly for both models.

From the results in Table 10, it can be observed that the performance of ImagenFew exhibits a significant decline after uniformly adopting the data partitioning and preprocessing strategy of `PreDiff`. For example, under the 10% data availability setting on the Stocks dataset, the ACD metric of ImagenFew sharply drops from 0.352 to 1.968, while on the ETTh dataset with 10% data availability, the ACD increases from 0.487 to 1.805. Moreover, the previously leading SD and KD metrics of ImagenFew show clear degradation, indicating that its advantage in statistical characteristics is substantially weakened under a fair comparison environment.

It is worth noting that this marked performance drop of ImagenFew confirms its dependence on specific data preprocessing methods. Furthermore, analysis reveals that, compared with the data partitioning and preprocessing strategy of ImagenFew, the design of `PreDiff` in this aspect is more reasonable and realistic: first, in the time series domain, sensor failures and malfunctions often lead to data collection containing only short continuous segments, which is a common scenario of data scarcity. Second, normalizing on the complete dataset before resampling implicitly introduces global statistical features, causing a distribution shift between randomly sampled data and real-world scarce

Table 12: Comprehensive comparison of ImagenFew and `PreDiff` on three datasets under varying data availability levels (100%, 70%, 40%, and 10%), with identical data partitioning and preprocessing procedures. Best results are highlighted in red.

| Dataset | Metrics | 100% | | 70% | | 40% | | 10% | |
|---|---|---|---|---|---|---|---|---|---|
| | | ImagenFew | PreDiff | ImagenFew | PreDiff | ImagenFew | PreDiff | ImagenFew | PreDiff |
| Energy | MDD↓ | **0.302** | 0.951 | **0.416** | 1.066 | **0.487** | 1.131 | **0.595** | 1.427 |
| | ACD↓ | 0.444 | **0.237** | 0.737 | **0.289** | 1.199 | **0.319** | 2.140 | **0.317** |
| | SD↓ | 0.425 | **0.308** | 0.560 | **0.372** | 0.495 | **0.446** | 0.975 | **0.652** |
| | KD↓ | 16.049 | **1.418** | 13.442 | **1.739** | 6.596 | **1.930** | 47.108 | **2.094** |
| | ED↓ | 1.099 | **1.048** | 1.179 | **1.151** | 1.346 | **1.329** | **1.588** | 1.722 |
| | DTW↓ | 6.635 | **6.549** | 7.046 | **6.821** | 7.869 | **6.949** | 9.128 | **7.216** |
| Stocks | MDD↓ | **0.287** | 0.356 | **0.529** | 0.797 | **0.552** | 1.158 | **0.873** | 1.724 |
| | ACD↓ | 0.115 | **0.051** | 0.065 | **0.088** | 0.481 | **0.414** | 1.968 | **0.728** |
| | SD↓ | **0.308** | 0.328 | 0.505 | **0.377** | 0.876 | **1.209** | 0.997 | 1.342 |
| | KD↓ | 2.016 | **1.598** | 2.431 | **1.805** | 2.946 | **1.033** | 2.444 | **1.629** |
| | ED↓ | 1.192 | **1.061** | 1.278 | **1.139** | 1.362 | **1.361** | 1.418 | 1.499 |
| | DTW↓ | 3.068 | **2.906** | 3.272 | **2.921** | 3.526 | **3.474** | 3.554 | 3.898 |
| ETTh | MDD↓ | **0.187** | 0.292 | **0.385** | 0.493 | 0.678 | **0.511** | 0.884 | **0.872** |
| | ACD↓ | 0.320 | **0.188** | 0.486 | **0.209** | 1.201 | **0.286** | 1.805 | **1.009** |
| | SD↓ | **0.163** | 0.206 | 0.557 | **0.213** | 0.804 | **0.239** | 1.010 | **0.369** |
| | KD↓ | **0.322** | 0.574 | 1.263 | **0.673** | 1.693 | **0.721** | 2.031 | **1.661** |
| | ED↓ | 0.892 | **0.823** | 0.969 | **0.870** | 1.292 | **0.908** | 1.427 | **1.081** |
| | DTW↓ | 2.513 | **2.269** | 2.721 | **2.475** | 3.682 | **2.549** | 4.035 | **3.049** |

scenarios. This makes it difficult to accurately simulate data scarcity conditions and, to some extent, undermines fairness.

# H  SIZE OF $X_{prior}$

We observed that the synthetic prior dataset $X_{prior}$, generated using ForecastPFN (Dooley et al., 2024), achieves strong performance. Therefore, we further investigate the impact of $X_{prior}$ size on the performance of `PreDiff`.

The results in Table 13 demonstrate that increasing the size of $X_{prior}$ generally improves the performance of `PreDiff`, as larger datasets provide more diverse and representative samples for pre-training. While larger datasets yield better performance, the marginal gains diminish as the dataset size grows beyond $100K$. This suggests that a dataset size of $100K$ may offer a good balance between performance and computational cost.

The best performance is achieved with the largest dataset size ($10M$), as evidenced by the lowest values in MDD, ED, and DTW. However, the $100K$ dataset size also performs exceptionally well, achieving the best results in ACD, SD, and KD. For real-world applications, selecting an appropriate $X_{prior}$ size depends on the specific requirements for performance and computational efficiency. A size of $100K$ may be sufficient for many scenarios, while larger sizes (e.g., $10M$) can be used when higher precision is critical.

This analysis highlights the importance of dataset size in the performance of **PreDiff** and provides practical guidance for selecting $X_{prior}$ sizes in time series generation tasks.

Table 13: The performance of `PreDiff` with varying sizes of the synthetic prior dataset $X_{prior} \in \{10K, 100K, 500K, 1M, 10M\}$ on the 10% of Stock dataset.

| Size | 10K | 100K | 500K | 1M | 10M |
|---|---|---|---|---|---|
| MDD | 2.151 | 1.724 | 1.701 | 1.715 | **1.697** |
| ACD | 0.967 | **0.728** | 0.785 | 0.737 | 0.736 |
| SD | 1.727 | **1.342** | 1.344 | 1.378 | 1.354 |
| KD | 2.082 | 1.629 | 1.685 | 1.641 | **1.624** |
| ED | 1.904 | 1.499 | 1.545 | **1.479** | 1.485 |
| DTW | 4.294 | 3.898 | 3.86 | 3.895 | **3.825** |

Table 14: Optimal $t_0$ values on different data.

| Dataset | $t_0/T$ | Data Size |
|---|---|---|
| Stock | 0.8 | 3294 |
| fMRI | 0.5 | 10000 |
| ETTh | 0.2 | 17421 |
| Energy | 0.2 | 19737 |

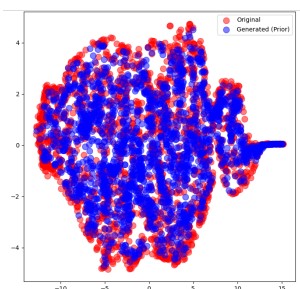 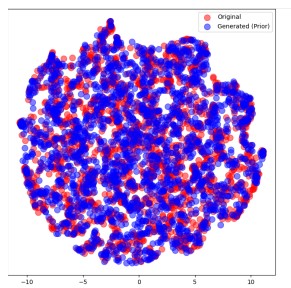 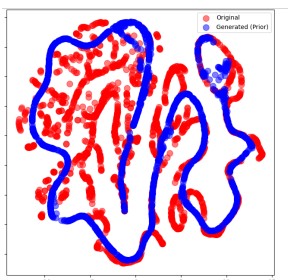

Figure 5: t-SNE visualization results across three datasets.

## I PARAMETERS ANALYSIS

The boundary step $t_0$ in `PreDiff` marks where pre-training on synthetic priors ends and fine-tuning on target data begins. Properly choosing $t_0$ balances the synthetic and real data distributions. We evaluated $t_0/T$ ratios of 20%, 50%, and 80% across multiple datasets and identified the optimal value for each. Table 14 shows that the best $t_0/T$ decreases as the target dataset size grows, reflecting a more diffusion steps are considered in pre-training when richer, larger real datasets better capture the true distribution.

Our investigation reveals that the generation quality of the model is highly sensitive to the choice of the splitting point $t_0$. Experiments conducted on the *Stocks* dataset show that optimal performance is achieved when $t_0/T = 0.6$, while deviations from this value lead to substantial degradation across multiple evaluation metrics. This sensitivity reflects the intrinsic difficulty of learning from datasets characterized by strong volatility, weak periodicity, and trend-dominated dynamics. For such datasets, setting $t_0$ to a relatively large value (e.g., $t_0/T \in [0.6, 0.8]$) ensures sufficient modeling of long-term dependencies before fine-grained tuning.

Furthermore, our decision to split the diffusion process into $[t_0, T]$ and $[0, t_0]$ is grounded in extensive empirical evidence and modeling intuition. Selecting an appropriate $t_0$ is crucial for mitigating overfitting and prior bias, ultimately shaping the quality of the generated sequences. To facilitate reproducibility, we offer the following guidelines: (1) **Dataset size**: larger datasets generally warrant smaller $t_0/T$; (2) **Learning difficulty**: datasets with strong periodicity (e.g., ETTh) benefit from smaller $t_0/T$, whereas highly volatile datasets (e.g., *Stocks*) require larger values.

In summary, for datasets with high feature-learning difficulty, we recommend initializing $t_0/T \in [0.6, 0.8]$; for datasets with lower difficulty or strong periodic structure, we recommend $t_0/T \in [0.2, 0.5]$, followed by local refinement. These findings underscore both the necessity and the effectiveness of dataset-specific tuning of the splitting point.

## J VISUALIZATIONS

In Figure 5, the t-SNE visualization results indicate that `PreDiff` consistently compresses the generated samples into nearly the same distributional region as the original data across three temporal scenarios. This overlap is consistently observed on the Energy, Sine, and Stock datasets. On the Stocks dataset, the generated samples almost perfectly align with the original ones, suggesting high fidelity without any visible mode collapse. On the Energy dataset, the generated samples align well with the centers of the original clusters, but exhibit a slightly larger cluster radius with diffuse boundary points, indicating high fidelity with increased diversity. On the Sine dataset, the generated samples follow the original "S"-shaped manifold, with only a slightly increased local thickness, demonstrating that `PreDiff` preserves the continuous phase–amplitude structure while introducing moderate interpolative

Overall, `PreDiff` achieves a desirable balance between high fidelity and moderate diversity. Furthermore, if sharper cluster boundaries or smoother manifold geometry are desired, one may simply fine-tune the prior variance or apply lightweight manifold regularization.

# K COMPARISON OF TWO PRETRAINING STRATEGIES ON IMAGENFEW

We pretrain ImagenFew using both its original pretraining strategy and the pretraining strategy of `PreDiff`, followed by fine-tuning under identical experimental settings. The resulting experimental outcomes are presented in Table 13.

Table 15: Comparison of ImagenFew under different pretraining strategies on the Stocks dataset with 10% data availability.

| Pretraining Strategy | ImagenFew w/ PreDiff Pretrain | ImagenFew w/ Original Pretrain |
|---|---|---|
| DS↓ | **0.478** | 0.4963 |
| PS↓ | **0.1429** | 0.1694 |
| C-FID↓ | **4.4694** | 5.8743 |
| MDD↓ | 1.0172 | **0.8404** |
| ACD↓ | 3.0636 | **2.2146** |
| SD↓ | **1.0429** | 1.0433 |
| KD↓ | **2.4053** | 2.4478 |
| ED↓ | **1.3426** | 1.5558 |
| DTW↓ | **3.3671** | 4.0789 |

As shown in Table 13, the performance of ImagenFew under different pretraining strategies is compared on the Stocks dataset with 10% data availability. The results indicate that the model pretrained with the `PreDiff` strategy outperforms the original pretraining strategy across multiple key metrics, particularly under the data-scarce (10%) setting. By leveraging synthetic data for pretraining and fine-tuning on target data, the `PreDiff` strategy effectively enhances the model's generalization ability and robustness, making its statistical distribution and autocorrelation more consistent with real data. This demonstrates that the `PreDiff` pretraining strategy holds significant advantages in generating high-quality time series data under data-scarcity conditions.

# L COMPLETE EXPERIMENTAL RESULTS

Table 16: The comprehensive comparison of PreDiff against state-of-the-art time series generation models on the Stock dataset at varying data availability levels (100%, 70%, 40%, and 10%) with the comprehensive experimental results across multiple random seeds. Red text denotes the best results, and blue text denotes the second-best.

| Models | Metric | TimeGAN | TimeVAE | DiffTime | DiffWave | FIDE | Diff-TS | PreDiff (Ours) |
|---|---|---|---|---|---|---|---|---|
| 100% | MDD↓ | 0.373±0.011 | 0.358±0.012 | 0.365±0.013 | 0.357±0.010 | 0.359±0.014 | 0.349±0.011 | 0.356±0.012 |
| | ACD↓ | 0.054±0.002 | 0.058±0.003 | 0.059±0.004 | 0.046±0.003 | 0.066±0.004 | 0.053±0.002 | 0.051±0.003 |
| | SD↓ | 0.359±0.011 | 0.335±0.010 | 0.336±0.010 | 0.340±0.012 | 0.345±0.013 | 0.322±0.010 | 0.328±0.010 |
| | KD↓ | 1.626±0.045 | 1.698±0.051 | 1.690±0.047 | 1.597±0.042 | 1.603±0.039 | 1.594±0.041 | 1.598±0.044 |
| | ED↓ | 1.093±0.032 | 1.116±0.038 | 1.104±0.033 | 1.111±0.036 | 1.105±0.031 | 1.087±0.030 | 1.061±0.028 |
| | DTW↓ | 2.899±0.095 | 2.910±0.102 | 2.901±0.097 | 2.896±0.099 | 2.901±0.091 | 2.897±0.094 | 2.906±0.086 |
| 70% | MDD↓ | 0.855±0.028 | 0.851±0.024 | 0.845±0.022 | 0.831±0.020 | 0.827±0.018 | 0.810±0.017 | 0.797±0.019 |
| | ACD↓ | 0.082±0.004 | 0.085±0.005 | 0.079±0.003 | 0.080±0.003 | 0.091±0.004 | 0.085±0.005 | 0.088±0.003 |
| | SD↓ | 0.403±0.017 | 0.402±0.018 | 0.405±0.018 | 0.387±0.016 | 0.395±0.017 | 0.386±0.016 | 0.377±0.014 |
| | KD↓ | 1.856±0.067 | 1.853±0.074 | 1.854±0.059 | 1.837±0.056 | 1.845±0.057 | 1.823±0.051 | 1.805±0.055 |
| | ED↓ | 1.145±0.039 | 1.151±0.043 | 1.160±0.044 | 1.156±0.042 | 1.187±0.040 | 1.145±0.038 | 1.139±0.036 |
| | DTW↓ | 2.978±0.103 | 2.983±0.111 | 2.976±0.099 | 2.964±0.096 | 2.958±0.088 | 2.943±0.089 | 2.921±0.092 |
| 40% | MDD↓ | 1.166±0.052 | 1.181±0.061 | 1.166±0.056 | 1.177±0.059 | 1.164±0.051 | 1.163±0.050 | 1.158±0.048 |
| | ACD↓ | 0.424±0.019 | 0.444±0.021 | 0.424±0.018 | 0.417±0.017 | 0.421±0.019 | 0.419±0.018 | 0.414±0.016 |
| | SD↓ | 1.217±0.054 | 1.213±0.060 | 1.218±0.058 | 1.206±0.052 | 1.215±0.057 | 1.206±0.050 | 1.209±0.051 |
| | KD↓ | 1.054±0.041 | 1.041±0.044 | 1.043±0.040 | 1.050±0.039 | 1.049±0.042 | 1.041±0.038 | 1.033±0.036 |
| | ED↓ | 1.393±0.052 | 1.375±0.055 | 1.379±0.053 | 1.371±0.051 | 1.386±0.052 | 1.364±0.047 | 1.362±0.046 |
| | DTW↓ | 3.498±0.128 | 3.502±0.132 | 3.505±0.131 | 3.497±0.129 | 3.499±0.127 | 3.489±0.125 | 3.474±0.121 |
| 10% | MDD↓ | 1.754±0.076 | 1.776±0.082 | 1.756±0.078 | 1.782±0.083 | 1.787±0.084 | 1.764±0.071 | 1.724±0.064 |
| | ACD↓ | 0.742±0.031 | 0.761±0.034 | 0.777±0.036 | 0.764±0.033 | 0.765±0.030 | 0.751±0.032 | 0.728±0.028 |
| | SD↓ | 1.471±0.061 | 1.477±0.067 | 1.471±0.059 | 1.470±0.056 | 1.476±0.058 | 1.455±0.054 | 1.342±0.050 |
| | KD↓ | 1.661±0.070 | 1.675±0.071 | 1.669±0.067 | 1.662±0.065 | 1.669±0.064 | 1.643±0.059 | 1.629±0.057 |
| | ED↓ | 1.514±0.058 | 1.560±0.062 | 1.506±0.057 | 1.525±0.060 | 1.532±0.061 | 1.511±0.059 | 1.499±0.051 |
| | DTW↓ | 3.930±0.138 | 3.957±0.147 | 3.913±0.141 | 3.924±0.142 | 3.921±0.144 | 3.905±0.130 | 3.898±0.123 |

Table 17: Comprehensive comparison of PreDiff against six baselines across four datasets under 10% data availability, with aggregated results over multiple random seeds (mean ± std). Lower values indicate better performance.

| Models | Metric | TimeGAN | TimeVAE | DiffTime | DiffWave | FIDE | Diff-TS | PreDiff (Ours) |
|---|---|---|---|---|---|---|---|---|
| Stock | MDD↓ | 1.754±0.034 | 1.776±0.036 | 1.766±0.041 | 1.782±0.035 | 1.787±0.038 | 1.764±0.027 | 1.724±0.029 |
| | ACD↓ | 0.742±0.014 | 0.761±0.012 | 0.777±0.016 | 0.764±0.010 | 0.765±0.017 | 0.751±0.013 | 0.728±0.011 |
| | SD↓ | 1.471±0.022 | 1.477±0.028 | 1.471±0.021 | 1.470±0.018 | 1.476±0.026 | 1.455±0.020 | 1.342±0.018 |
| | KD↓ | 1.661±0.041 | 1.675±0.043 | 1.669±0.037 | 1.662±0.033 | 1.669±0.044 | 1.643±0.035 | 1.629±0.034 |
| | ED↓ | 1.514±0.025 | 1.560±0.021 | 1.506±0.021 | 1.525±0.021 | 1.532±0.024 | 1.511±0.020 | 1.499±0.018 |
| | DTW↓ | 3.930±0.071 | 3.957±0.083 | 3.913±0.059 | 3.924±0.055 | 3.921±0.066 | 3.905±0.048 | 3.898±0.041 |
| fMRI | MDD↓ | 0.558±0.018 | 0.552±0.019 | 0.564±0.017 | 0.563±0.020 | 0.555±0.021 | 0.547±0.015 | 0.524±0.014 |
| | ACD↓ | 0.635±0.013 | 0.626±0.011 | 0.627±0.010 | 0.645±0.019 | 0.639±0.010 | 0.625±0.009 | 0.608±0.008 |
| | SD↓ | 0.755±0.020 | 0.755±0.022 | 0.744±0.018 | 0.759±0.023 | 0.760±0.019 | 0.738±0.017 | 0.729±0.014 |
| | KD↓ | 0.431±0.012 | 0.440±0.015 | 0.419±0.012 | 0.409±0.011 | 0.421±0.013 | 0.421±0.014 | 0.410±0.010 |
| | ED↓ | 1.632±0.030 | 1.634±0.028 | 1.652±0.031 | 1.642±0.032 | 1.642±0.027 | 1.635±0.021 | 1.609±0.019 |
| | DTW↓ | 7.248±0.084 | 7.239±0.078 | 7.245±0.072 | 7.249±0.068 | 7.252±0.070 | 7.238±0.065 | 7.222±0.058 |
| ETTh | MDD↓ | 0.885±0.019 | 0.889±0.018 | 0.902±0.023 | 0.885±0.017 | 0.905±0.025 | 0.889±0.017 | 0.872±0.014 |
| | ACD↓ | 1.036±0.024 | 1.033±0.022 | 1.038±0.023 | 1.019±0.021 | 1.024±0.018 | 1.027±0.020 | 1.009±0.016 |
| | SD↓ | 0.372±0.011 | 0.362±0.009 | 0.358±0.010 | 0.380±0.014 | 0.353±0.009 | 0.362±0.010 | 0.369±0.013 |
| | KD↓ | 1.674±0.030 | 1.668±0.027 | 1.677±0.032 | 1.668±0.028 | 1.673±0.031 | 1.671±0.025 | 1.661±0.022 |
| | ED↓ | 1.098±0.026 | 1.073±0.020 | 1.091±0.024 | 1.074±0.023 | 1.089±0.024 | 1.091±0.018 | 1.081±0.016 |
| | DTW↓ | 3.066±0.047 | 3.057±0.033 | 3.070±0.035 | 3.065±0.039 | 3.074±0.042 | 3.064±0.036 | 3.049±0.030 |
| Energy | MDD↓ | 1.470±0.026 | 1.462±0.025 | 1.461±0.022 | 1.469±0.029 | 1.454±0.020 | 1.456±0.021 | 1.427±0.017 |
| | ACD↓ | 0.367±0.012 | 0.365±0.013 | 0.365±0.012 | 0.381±0.015 | 0.361±0.011 | 0.352±0.010 | 0.317±0.009 |
| | SD↓ | 0.671±0.021 | 0.669±0.022 | 0.669±0.020 | 0.675±0.022 | 0.649±0.017 | 0.655±0.019 | 0.652±0.016 |
| | KD↓ | 2.110±0.045 | 2.105±0.042 | 2.099±0.040 | 2.105±0.043 | 2.127±0.052 | 2.106±0.039 | 2.094±0.036 |
| | ED↓ | 1.741±0.028 | 1.742±0.032 | 1.730±0.023 | 1.739±0.025 | 1.743±0.024 | 1.738±0.019 | 1.722±0.017 |
| | DTW↓ | 7.230±0.081 | 7.235±0.084 | 7.232±0.077 | 7.217±0.066 | 7.234±0.089 | 7.228±0.076 | 7.209±0.061 |

Table 18: Comprehensive comparison of PreDiff and Diff-TS across four datasets under varying data availability levels, with aggregated results over multiple random seeds (values in parentheses are standard deviations).

| Per(%) | | 100% | | 70% | | 40% | | 10% | |
|---|---|---|---|---|---|---|---|---|---|
| Models | | Diff-TS | **PreDiff** | Diff-TS | **PreDiff** | Diff-TS | **PreDiff** | Diff-TS | **PreDiff** |
| Stock | MDD↓ | **0.349 (0.008)** | 0.356 (0.007) | 0.810 (0.015) | **0.797 (0.014)** | 1.163 (0.022) | **1.158 (0.020)** | 1.764 (0.035) | **1.724 (0.030)** |
| | ACD↓ | 0.053 (0.002) | **0.051 (0.002)** | **0.085 (0.003)** | 0.088 (0.004) | 0.419 (0.010) | **0.414 (0.009)** | 0.751 (0.015) | **0.728 (0.013)** |
| | SD↓ | **0.322 (0.006)** | 0.328 (0.007) | 0.386 (0.008) | **0.377 (0.007)** | **1.206 (0.025)** | 1.209 (0.024) | 1.455 (0.030) | **1.342 (0.028)** |
| | KD↓ | **1.594 (0.032)** | 1.598 (0.030) | 1.823 (0.035) | **1.805 (0.033)** | 1.041 (0.020) | **1.033 (0.019)** | 1.643 (0.028) | **1.629 (0.027)** |
| | ED↓ | 1.087 (0.022) | **1.061 (0.021)** | 1.145 (0.024) | **1.139 (0.022)** | 1.364 (0.028) | **1.362 (0.027)** | 1.511 (0.030) | **1.499 (0.029)** |
| | DTW↓ | **2.897 (0.058)** | 2.906 (0.056) | 2.943 (0.059) | **2.921 (0.058)** | 3.489 (0.070) | **3.474 (0.068)** | 3.905 (0.078) | **3.898 (0.076)** |
| fMRI | MDD↓ | 0.294 (0.006) | **0.288 (0.005)** | 0.311 (0.007) | **0.305 (0.006)** | 0.392 (0.009) | **0.374 (0.008)** | 0.547 (0.012) | **0.524 (0.011)** |
| | ACD↓ | 0.198 (0.004) | **0.194 (0.004)** | 0.239 (0.005) | **0.202 (0.004)** | 0.467 (0.010) | **0.432 (0.009)** | 0.625 (0.013) | **0.608 (0.012)** |
| | SD↓ | 0.154 (0.003) | **0.137 (0.003)** | 0.277 (0.006) | **0.271 (0.006)** | 0.429 (0.009) | **0.401 (0.008)** | 0.738 (0.015) | **0.729 (0.014)** |
| | KD↓ | 0.118 (0.002) | **0.106 (0.002)** | **0.153 (0.003)** | 0.159 (0.003) | 0.282 (0.006) | **0.258 (0.005)** | 0.421 (0.008) | **0.410 (0.008)** |
| | ED↓ | **0.873 (0.018)** | 0.877 (0.017) | 0.938 (0.019) | **0.932 (0.018)** | 1.236 (0.025) | **1.202 (0.022)** | 1.635 (0.030) | **1.609 (0.028)** |
| | DTW↓ | 6.088 (0.120) | **6.071 (0.118)** | 6.216 (0.123) | **6.201 (0.121)** | 6.623 (0.132) | **6.594 (0.130)** | 7.238 (0.145) | **7.222 (0.142)** |
| ETTh | MDD↓ | 0.307 (0.006) | **0.292 (0.005)** | 0.508 (0.010) | **0.493 (0.009)** | 0.529 (0.011) | **0.511 (0.010)** | 0.889 (0.018) | **0.872 (0.017)** |
| | ACD↓ | 0.201 (0.004) | **0.188 (0.004)** | 0.227 (0.005) | **0.209 (0.004)** | 0.295 (0.006) | **0.286 (0.006)** | 1.027 (0.020) | **1.009 (0.019)** |
| | SD↓ | 0.217 (0.005) | **0.206 (0.004)** | 0.225 (0.005) | **0.213 (0.005)** | 0.257 (0.006) | **0.239 (0.005)** | **0.362 (0.008)** | 0.369 (0.008) |
| | KD↓ | 0.598 (0.012) | **0.574 (0.011)** | **0.664 (0.013)** | 0.673 (0.013) | 0.739 (0.015) | **0.721 (0.014)** | 1.671 (0.034) | **1.661 (0.033)** |
| | ED↓ | **0.821 (0.016)** | 0.823 (0.016) | 0.902 (0.018) | **0.870 (0.017)** | 0.917 (0.018) | **0.908 (0.017)** | 1.091 (0.022) | **1.081 (0.022)** |
| | DTW↓ | 2.288 (0.046) | **2.269 (0.045)** | 2.503 (0.050) | **2.475 (0.048)** | 2.554 (0.051) | **2.549 (0.051)** | 3.064 (0.061) | **3.049 (0.060)** |
| Energy | MDD↓ | 0.960 (0.019) | **0.951 (0.018)** | 1.068 (0.021) | **1.066 (0.021)** | 1.139 (0.023) | **1.131 (0.022)** | 1.456 (0.029) | **1.427 (0.028)** |
| | ACD↓ | 0.243 (0.005) | **0.237 (0.005)** | **0.287 (0.006)** | 0.289 (0.006) | 0.327 (0.007) | **0.319 (0.007)** | 0.352 (0.008) | **0.317 (0.007)** |
| | SD↓ | 0.324 (0.007) | **0.308 (0.007)** | 0.378 (0.008) | **0.372 (0.008)** | 0.463 (0.010) | **0.446 (0.009)** | 0.655 (0.013) | **0.652 (0.012)** |
| | KD↓ | 1.431 (0.029) | **1.418 (0.028)** | 1.756 (0.035) | **1.739 (0.034)** | 1.969 (0.039) | **1.930 (0.038)** | 2.106 (0.042) | **2.094 (0.041)** |
| | ED↓ | **1.046 (0.021)** | 1.048 (0.021) | 1.153 (0.023) | **1.151 (0.022)** | 1.346 (0.026) | **1.329 (0.025)** | 1.738 (0.034) | **1.722 (0.033)** |
| | DTW↓ | **6.541 (0.131)** | 6.549 (0.130) | 6.825 (0.137) | **6.821 (0.136)** | 6.952 (0.139) | **6.949 (0.138)** | 7.228 (0.145) | **7.216 (0.144)** |

Table 19: Classification accuracy in EEG Eye dataset (mean ± std). Higher values indicate better performance.

| Metric | TimeGAN | TimeVAE | DiffTime | DiffWave | FIDE | Diff-TS | PreDiff |
|---|---|---|---|---|---|---|---|
| Accuracy | 0.529±0.012 | 0.543±0.011 | 0.601±0.010 | 0.587±0.010 | 0.591±0.009 | 0.605±0.008 | 0.622±0.007 |

Table 20: Comprehensive results (mean ± std) of `PreDiff` on different $X_{prior}$ datasets with $100\,K$ and $10\,M$ data aggregated over multiple random seeds; lower is better.

| Size | Metric | ForecastPFN | SNIP | Monash |
|---|---|---|---|---|
| $100K$ | MDD | 1.724±0.029 | 2.016±0.034 | **1.718±0.028** |
| | ACD | **0.728±0.011** | 0.754±0.012 | 0.832±0.014 |
| | SD | **1.342±0.018** | 1.359±0.020 | 1.351±0.019 |
| | KD | **1.629±0.034** | 1.763±0.037 | 1.922±0.042 |
| | ED | **1.499±0.018** | 1.636±0.021 | 1.526±0.020 |
| | DTW | 3.898±0.041 | 3.973±0.048 | **3.872±0.039** |
| $10M$ | MDD | **1.697±0.026** | 2.001±0.033 | 1.736±0.031 |
| | ACD | **0.736±0.010** | 0.772±0.012 | 0.851±0.013 |
| | SD | 1.354±0.019 | **1.327±0.017** | 1.377±0.020 |
| | KD | **1.624±0.032** | 1.729±0.036 | 1.939±0.041 |
| | ED | **1.485±0.017** | 1.685±0.022 | 1.511±0.019 |
| | DTW | **3.825±0.038** | 3.985±0.045 | 3.872±0.041 |

Table 21: Comprehensive comparison of ImagenFew against `PreDiff` across three datasets under varying data availability levels (100%, 70%, 40%, and 10%). Best results are highlighted in red. Mean ± std over multiple runs.

| Dataset | Metrics | 100% | | 70% | | 40% | | 10% | |
|---------|---------|----------|---------|----------|---------|----------|---------|----------|---------|
| | | ImagenFew | PreDiff | ImagenFew | PreDiff | ImagenFew | PreDiff | ImagenFew | PreDiff |
| Energy | MDD↓ | **0.314±0.012** | 0.951±0.028 | **0.339±0.014** | 1.066±0.031 | **0.399±0.015** | 1.131±0.033 | **0.487±0.018** | 1.427±0.037 |
| | ACD↓ | 0.444±0.010 | **0.237±0.008** | 0.738±0.012 | **0.289±0.009** | 1.281±0.015 | **0.319±0.010** | 2.256±0.020 | **0.317±0.011** |
| | SD↓ | 0.425±0.011 | **0.308±0.009** | 0.527±0.012 | **0.372±0.010** | 0.526±0.011 | **0.446±0.010** | 1.504±0.022 | **0.652±0.013** |
| | KD↓ | 12.953±0.032 | **1.418±0.028** | 21.889±0.045 | **1.739±0.030** | 17.901±0.038 | **1.930±0.031** | 151.784±0.072 | **2.094±0.034** |
| | ED↓ | 1.094±0.020 | **1.048±0.018** | 1.146±0.022 | **1.151±0.019** | 1.253±0.024 | **1.329±0.020** | 1.462±0.027 | **1.722±0.022** |
| | DTW↓ | 6.610±0.035 | **6.549±0.033** | 6.872±0.038 | **6.821±0.035** | 7.346±0.041 | **6.949±0.037** | 8.447±0.050 | **7.216±0.040** |
| Stocks | MDD↓ | **0.288±0.010** | 0.356±0.012 | **0.318±0.012** | 0.797±0.029 | **0.365±0.013** | 1.158±0.031 | **0.401±0.015** | 1.724±0.033 |
| | ACD↓ | 0.116±0.005 | **0.051±0.004** | 0.114±0.005 | **0.088±0.006** | 0.251±0.008 | **0.414±0.010** | **0.352±0.009** | 0.728±0.011 |
| | SD↓ | **0.310±0.008** | 0.328±0.009 | **0.372±0.010** | 0.377±0.011 | **0.447±0.012** | 1.209±0.028 | **0.488±0.011** | 1.342±0.020 |
| | KD↓ | 2.020±0.015 | **1.598±0.012** | 2.236±0.018 | **1.805±0.014** | 2.601±0.020 | **1.033±0.010** | 2.681±0.022 | **1.629±0.013** |
| | ED↓ | 1.189±0.016 | **1.061±0.012** | 1.179±0.015 | **1.139±0.014** | 1.212±0.017 | **1.362±0.013** | 1.174±0.014 | **1.499±0.011** |
| | DTW↓ | 3.060±0.020 | **2.906±0.018** | 3.031±0.021 | **2.921±0.019** | 3.126±0.022 | **3.474±0.020** | 3.055±0.021 | **3.898±0.019** |
| ETTh | MDD↓ | **0.023±0.001** | 0.292±0.010 | **0.024±0.001** | 0.493±0.015 | **0.027±0.002** | 0.511±0.018 | **0.031±0.002** | 0.872±0.025 |
| | ACD↓ | 0.244±0.008 | **0.188±0.007** | 0.287±0.010 | **0.209±0.008** | 0.358±0.012 | **0.286±0.009** | 0.487±0.015 | **1.009±0.012** |
| | SD↓ | **0.085±0.004** | 0.206±0.009 | **0.026±0.003** | 0.213±0.010 | **0.079±0.004** | 0.239±0.011 | **0.164±0.006** | 0.369±0.012 |
| | KD↓ | **0.364±0.014** | 0.574±0.018 | **0.149±0.010** | 0.673±0.020 | **0.383±0.015** | 0.721±0.022 | **0.186±0.011** | 1.661±0.028 |
| | ED↓ | 6.026±0.035 | **0.823±0.019** | 6.094±0.038 | **0.870±0.021** | 6.078±0.040 | **0.908±0.022** | 6.148±0.042 | **1.081±0.018** |
| | DTW↓ | 16.747±0.050 | **2.269±0.025** | 16.970±0.055 | **2.475±0.028** | 16.915±0.058 | **2.549±0.030** | 17.058±0.060 | **3.049±0.033** |

Table 22: Sensitivity of Generation Quality to the Pre-training–Fine-tuning Cut-off Step $t_0/T$ on the Stocks Dataset (mean ± std).

| $t_0/T$ | MDD | ACD | SD | KD | ED | DTW |
|---------|-----|-----|----|----|----|----|
| 0.1 | 1.4953±0.012 | 4.6021±0.035 | 0.5013±0.010 | 2.2559±0.020 | 1.3189±0.018 | 3.3668±0.022 |
| 0.2 | 1.4979±0.013 | 3.6204±0.025 | 0.5376±0.011 | 2.3769±0.021 | 1.3674±0.019 | 3.5219±0.024 |
| 0.3 | 1.5995±0.015 | 4.6938±0.038 | 0.7198±0.014 | 2.6055±0.022 | 1.3692±0.020 | 3.4928±0.023 |
| 0.4 | 1.4463±0.011 | 3.1264±0.021 | 0.7584±0.014 | 2.7387±0.024 | 1.2130±0.016 | 3.1588±0.020 |
| 0.5 | 1.5612±0.014 | 3.6939±0.026 | 0.5974±0.012 | 2.2831±0.021 | 1.2801±0.017 | 3.3264±0.022 |
| 0.6 | 1.3992±0.010 | 3.1089±0.020 | 0.3948±0.008 | 2.4328±0.019 | 1.3934±0.018 | 3.6810±0.024 |
| 0.7 | 1.7664±0.016 | 4.0166±0.031 | 0.6497±0.013 | 2.2884±0.020 | 1.4189±0.019 | 3.6948±0.025 |
| 0.8 | 1.5603±0.014 | 3.4891±0.025 | 0.7104±0.014 | 2.2442±0.018 | 1.3548±0.018 | 3.5688±0.023 |
| 0.9 | 1.9687±0.018 | 4.8139±0.040 | 0.7991±0.015 | 2.4227±0.020 | 1.1439±0.010 | 2.9169±0.019 |
| 0.999 | 1.8266±0.017 | 4.6018±0.035 | 0.6569±0.013 | 2.3816±0.021 | 1.1549±0.011 | 2.8831±0.018 |

