# OpenReview forum: "PreDiff: Leveraging Data Priors to Enhance Time Series Generation with Scarce Samples"
_ICLR.cc/2026/Conference — Submitted to ICLR 2026_

### Official Review · Reviewer_uvZ5 · 2025-10-25

**Soundness:** 2
**Presentation:** 2
**Contribution:** 2
**Rating:** 4
**Confidence:** 3

**Summary:**

This paper studies time series generation models, particularly diffusion models, which suffer performance degradation when trained on scarce data. To address this, the authors propose PreDiff, a two-stage training framework. The first stage pre-trains a diffusion model on a large synthetic prior dataset), on the latter half of the denoising process. The second stage fine-tunes the model on the small target dataset), on the initial half of the process, with the parameters from the first stage frozen. The authors claim this method effectively leverages general priors to mitigate overfitting and achieves state-of-the-art results.

**Strengths:**

1. **Novel Training Heuristic**: The core idea of splitting the diffusion process at a point t0​ for pre-training and fine-tuning is a novel and clever heuristic. The intuition of learning varying time series data structure from a large prior and, then learning details from the target data is conceptually sound.

2. **Strong Empirical Results**: The ablation in Figure 3(a) is the most compelling part of the paper. It clearly demonstrates that the proposed two-stage split method outperforms simpler alternatives like "Pre-training Only," "Fine-tuning Only," and "Datasets Mixing," which validates the efficacy of the proposed training strategy over these baselines. Also, the tables 1 and 2 show that the proposed algorithm outperforms several baseline algorithms.

**Weaknesses:**

1. **Some Results Weakening the Motivation**: The paper's core premise is to resolve data scarcity. However, the authors state in Section 5.5 (and show in Table 5) that the method performs well when the prior data is close to the target data. This implies that one must already have access to a large, well-matched dataset or a synthetic generator that knows the target's core properties. I think this is a form of transfer learning from a known, similar source, which somewhat contradicts the "data scarcity" scenario where such well-matched priors are, by definition, unavailable.

2. **Lack of Technical Rigor and Theoretical Grounding**:
 - In Section 3, the paper defines $\mu_\theta$​ as the model predicting the mean of the reverse process. However, the loss functions $L_{\text{pre}}$​ (Eq. 2) and $L_{\text{ft}}$​ (Eq. 3) train this model to predict the noise $\epsilon$ (i.e., $|| \mu_\theta (\cdot) -\epsilon||^2$. This is a bit confusion between the mean-predictor $\mu_\theta​$ and the noise-predictor $\epsilon_\theta$​, demonstrating a lack of technical precision.

 - Un-grounded Heuristic: The main contribution, the $t_0$​ split, is presented as an ad-hoc heuristic without any theoretical reasons. The paper claims $[t_0​,T]$ maps to "coarse structure" and $[0,t_0]$ to "fine details" but provides zero evidence for this assertion.

**Questions:**

1. Your results in Table 5 show that the prior seems to be "highly relevant" to the target for the proposed method to perform well. How do you argue this requirement with the paper's "data scarcity" motivation? I think a true scarce-data scenario implies such a large, well-matched prior is not available.

2. Can you please clarify the critical inconsistency in your method? Section 3 defines $\mu_\theta$​ as the mean-predictor (denoising process), but Equations 2 and 3 train it to target $\epsilon$​. Which is it? Also, please add $\epsilon$ to the expectation.

3. The core claim is that $[t_0, T]$ learns "coarse" priors and $[0, t_0]$ learns "fine" details. What theoretical or empirical evidence (e.g., visualizations of samples at step $t_0$​) can you provide to support this assertion? Without this, I think $t_0$​ split appears to be just an un-grounded, dataset-specific hyperparameter.

---

> ### Author Response · Authors · 2025-11-22
>
> Dear Reviewer,
>
> We have carefully revised the entire manuscript according to your comments and have uploaded the updated version. All modifications are highlighted in blue for easy reference. In addition, we have provided point-by-point responses to each of your comments. We hope that these revisions and explanations adequately address your concerns. If you have any further questions or suggestions, we would be very happy to continue the discussion.
>
> ### **Response to Weakness 1**
>
> We acknowledge that part of our method’s effectiveness comes from the quality of the constructed external priors, which are essential in low-sample settings. Without priors closely matching the target distribution, leveraging prior knowledge to improve generation quality would be impossible.
>
> The key innovation lies in synthesizing diverse priors to address data scarcity. These priors provide rich, generalizable features that compensate for limited diversity in low-data scenarios. Existing models cannot effectively utilize such priors, as verified in Appendix F (Table 10), where integrating our priors into DDPM yields suboptimal results compared to PreDiff.
>
> The priors are designed to work with our model architecture; they are mutually reinforcing. PreDiff enables zero-shot time series generation in data-scarce settings, demonstrating both the novelty and practical significance of our approach.
>
>
> We also attempt to investigate how different data distributions affect model performance by analyzing visualization results that reflect representation coverage, as described below.
> We quantify the range of representations covered by the datasets generated by ForecastPFN and Monash using a set of statistical metrics, including stationarity (ADF test), forecastability, frequency-domain characteristics (FFT mean), seasonality, trend (Mann–Kendall test), and permutation entropy (see Appendix F for full descriptions).
> We then use Radviz to visualize the high-dimensional statistical features of 256-length time-series segments drawn from both our synthetic data and the Monash datasets. From Monash—which includes weather, traffic, electricity, tourism, medical, and energy domains—we sample 100K segments per domain.
> The Radviz visualization (see Figure 4) shows that the distributional diversity of our synthetic dataset is higher than that of the Monash datasets. In addition, Table 5 demonstrates that pretraining on our synthetic prior data yields better generation quality than pretraining on Monash, providing further evidence of the higher quality of our synthesized priors.
>
>
> ### **Response to Weakness 2**
>
> $\mu_\theta$ should be $\epsilon_\theta$, and we have revise it in the revised version.
>
>
>
> ### **Response to Weakness 3**
>
> We thank the reviewers for their question regarding the choice of the splitting point $t_0$. Our experiments (Table 6 in the revised version) show that the generation quality is highly sensitive to $t_0$. On the Stocks dataset, the best performance occurs at $t_0/T=0.6$, and even small deviations cause notable performance drops. This dataset is highly volatile and weakly periodic, making feature learning difficult; thus, a larger $t_0/T$ is necessary to capture long-term dependencies.
>
> Our decision to divide the diffusion process into $[t_0,T]$ and $[0,t_0]$ is based on empirical validation. The choice of $t_0$ is crucial for avoiding overfitting and prior bias. To support reproducibility, we provide clear guidelines:
> – Use larger $t_0/T$ (0.6–0.8) for datasets with high volatility and learning difficulty;
> – Use smaller $t_0/T$ (0.2–0.5) for datasets with strong periodicity or simpler structure.
>
> These results demonstrate that $t_0$ must be tuned according to dataset characteristics and that our methodology provides stable, predictable improvements.
>
>
> ### **Response to Questions 1-3**
>
> Please see **Response to Weaknesses 1-3**.

---

> > ### Comment · Reviewer_uvZ5 · 2025-11-26
> >
> > I would like to thank authors for their responses and the revised manuscript.
> > I carefully read the updates, and decided to keep my original score unchanged.
> > The new experiments are helpful. However, the proposed method still feels somewhat heuristic.
> > The code is not publicly available, so reproducibility remains a concern. Best of luck with the review process.

---

> > > ### Author Response · Authors · 2025-11-28
> > >
> > > Thank you very much for taking the time again to review our response and revised manuscript, and we sincerely appreciate the insightful feedback. We provide a further response below:
> > >
> > > ### **Response regarding the code issue**
> > >
> > > Thank you for your valuable comments. Following your suggestions, we have been continuously organizing the code. To ensure reproducibility of experimental results, we have compiled and uploaded the full code, data preprocessing scripts, and training/evaluation instructions to the Supplementary Material, including:
> > >
> > > 1. Environment dependency file `requirements.txt`
> > > 2. Hyperparameter configurations `configs/`
> > > 3. complete pretraining, finetuning, sampling, and evaluation pipeline on the 10% Stocks dataset in `pre_DiffusionTS.ipynb`
> > >
> > > We have anonymized all identifying information. The reviewer may directly extract and run to reproduce results. Please feel free to contact us if further clarification is needed.
> > >
> > > ### **Question: The proposed method still feels somewhat heuristic.**
> > >
> > > First, thank you for your recognition of our work. We fully understand your concern regarding “the method still having heuristic characteristics.” Indeed, we currently lack more rigorous theoretical support or a unified framework, which represents a common challenge in this research area. We plan to strengthen this aspect in future work, including exploring more systematic theoretical foundations and more comprehensive analysis.
> > >
> > > Although heuristic elements exist, extensive experiments demonstrate that using the ( $t_0$ ) point as a split within the diffusion process for separate pretraining and finetuning is effective in improving generation quality under data-scarce scenarios (shown in Table 1 and Table 2 in the paper). This indicates that our method successfully addresses the severe degradation in generation quality faced by existing models under data scarcity. This evidence confirms that our work is meaningful and novel.
> > >
> > > Thank you again for the valuable feedback throughout the review process. It has been extremely helpful for guiding future improvements, and we warmly welcome further discussion and suggestions from you.

---

### Official Review · Reviewer_G8K1 · 2025-10-26

**Soundness:** 2
**Presentation:** 3
**Contribution:** 2
**Rating:** 4
**Confidence:** 4

**Summary:**

This paper addresses performance degradation in diffusion-based time series generation models under data scarcity. It proposes PreDiff, a two-stage framework: pre-training on a large data prior, synthetic or real, and fine-tuning on the target scarce dataset. Its claimed novelty is a specific "split-step" training strategy: pre-training focuses on later diffusion steps t_0 to T, global structure, while fine-tuning exclusively updates earlier steps 0 to t_0, fine details. Experiments aim to show improved generation quality under scarce data conditions.

**Strengths:**

1 Addresses the critical problem of TSG under data scarcity. The split-step training strategy, linking diffusion stages to transfer learning, is a conceptually distinct idea within this context.

2 Motivation is clear. The two-stage framework is presented logically. Experiments use standard benchmarks and show empirical benefits, particularly in severe data scarcity 10% data.

**Weaknesses:**

1 The central assumption linking diffusion steps to transferable features lacks strong theoretical support or broad empirical validation across diverse TS types/diffusion models presented here. Its effectiveness may be context-dependent.

2 Success hinges on a relevant, high-quality data prior X_prior. The paper offers little practical guidance on selecting or assessing prior suitability, posing a major barrier to reliable application and risking negative transfer.

3 Performance is likely sensitive to the split point t_0, yet guidance on its selection is minimal beyond empirical observation(Appendix I) . Lack of a principled selection method adds significant tuning difficulty.

4 As the pre-train/fine-tune paradigm is standard, the overall contribution relies heavily on the split-step strategy. If its universality is questionable, the novelty might be seen as incremental.

5 Comparison with ImagenFew highlights sensitivity to preprocessing, potentially affecting fairness and conclusions.

6 Reliance on potentially massive priors implies significant computational costs, limiting practicality, especially if priors need tailoring per domain.

**Questions:**

1 Can you provide stronger theoretical arguments or broader empirical evidence (e.g., across diverse datasets, different diffusion models) to support the universality of the hypothesis that pre-training high-noise steps and fine-tuning low-noise steps is an optimal transfer strategy for diffusion models?

2 How can a practitioner reliably select an effective data prior X_prior for a given scarce target dataset X_target? What happens if a truly relevant large prior is unavailable? Please elaborate on the risk and mitigation of negative transfer.

3 Given its likely sensitivity, how should t_0 be chosen in practice? Is there a risk that optimal t_0 heavily depends on the specific prior-target pair, requiring extensive tuning for each new application?

4 Considering the need for a massive relevant prior, the cost of pre-training, and the tuning required for t_0, how practical is PreDiff for real-world users facing data scarcity?

5 Could you clarify the impact of preprocessing differences and potentially provide results comparing PreDiff and ImagenFew under identical preprocessing settings to ensure fairness?

---

> ### Author Response · Authors · 2025-11-22
> **Response to Weaknesses 1–3**
>
> ### **Response to Weakness 1**
>
> We have added results using DDPM, DiffTime, and Diff-TS as baseline diffusion models for our method on the Stock, fMRI, ETTh, and Energy datasets (see Appendix F, Table 10 in the revised manuscript). The results demonstrate the flexibility of our proposed method.
>
>
> ### **Response to Weakness 2**
>
> 1) If relevant, high-quality datasets are available, we would naturally use them, which would likely lead to better generation results. However, due to the lack of readily available highly relevant priors, we opted to use a synthetic prior dataset to enhance prior knowledge. This choice represents a default setting.
>
> 2) Assessing the suitability of prior data is crucial. To address this, we designed experiments to evaluate the quality of the prior data, with visualization results as follows:
>
> We quantify the range of representations covered by datasets generated by ForecastPFN and Monash using various statistical metrics, including stationarity (ADF test), forecastability, frequency-domain characteristics (FFT mean), seasonality, trend (Mann–Kendall test), and permutation entropy (see Appendix F for full descriptions).
>
> We then apply Radviz to visualize the high-dimensional statistical features of 256-length time-series segments sampled from both our synthetic data and the Monash datasets. From Monash—which spans weather, traffic, electricity, tourism, medical, and energy domains—we sample 100K segments per domain.
>
> The Radviz visualization (Figure 4) shows that the distributional diversity of our synthetic dataset is higher than that of the Monash datasets. Moreover, Table 5 demonstrates that pretraining on our synthetic prior data achieves better generation quality than pretraining on Monash, further confirming the superior quality of our synthesized priors.
>
>
> ### **Response to Weakness 3**
>
> We thank the reviewers for their question regarding the choice of the splitting point $t_0$. Our experiments (Table 6 in the revised version) show that the generation quality is highly sensitive to $t_0$. On the Stocks dataset, the best performance occurs at $t_0/T=0.6$, and even small deviations cause notable performance drops. This dataset is highly volatile and weakly periodic, making feature learning difficult; thus, a larger $t_0/T$ is necessary to capture long-term dependencies.
>
> Our decision to divide the diffusion process into $[t_0,T]$ and $[0,t_0]$ is based on empirical validation. The choice of $t_0$ is crucial for avoiding overfitting and prior bias. To support reproducibility, we provide clear guidelines:
> – Use larger $t_0/T$ (0.6–0.8) for datasets with high volatility and learning difficulty;
> – Use smaller $t_0/T$ (0.2–0.5) for datasets with strong periodicity or simpler structure.
>
> These results demonstrate that $t_0$ must be tuned according to dataset characteristics and that our methodology provides stable, predictable improvements.

---

> ### Author Response · Authors · 2025-11-22
> **Response to Weaknesses 4-6 and Questions 1-5**
>
> ### **Response to Weakness 4**
>
> We agree that our approach is conceptually related to the general idea of “pretrained diffusion + fine-tuning,” which also appears in vision and text domains. However, applying this paradigm to time-series generation is nontrivial, and our method introduces several essential, domain-specific innovations that go far beyond a direct adaptation.
>
> **1. Why this framework is necessary for time-series generation**
> In many real-world scenarios, time-series data are extremely scarce due to privacy constraints, high collection costs, or the rarity of certain events. Existing diffusion-based generators rely heavily on large, high-quality datasets; as shown in Figure 1 of the paper, their performance deteriorates significantly under data scarcity. Our approach leverages diverse synthetic priors to compensate for this lack of data, enabling meaningful zero-shot time-series generation—an ability that genuinely matters for time-series applications.
>
> **2. Why this is not equivalent to vision/text “pretrain + fine-tune”**
> Although similar in high-level concept, our method requires several fundamental changes to accommodate the unique characteristics of time-series data:
>
> * **Diffusion-step decomposition.**
>   Instead of end-to-end pretraining as used in vision/text, we structurally split the diffusion trajectory:
>   – Pretraining focuses only on the denoising segment from an intermediate step ( $t_0$ ) to ( $T$ ), learning long-term structure and global dynamics from large-scale synthetic priors.
>   – Fine-tuning learns the early steps ( $0 \to t_0$ ), adapting localized patterns and short-term behaviors of the target dataset.
>   This explicit temporal decomposition directly addresses the coexistence of long-range dependencies and short-term fluctuations in time series—an issue that does not arise in vision or NLP.
>
> * **Domain-specific prior construction.**
>   Unlike ImageNet or web-scale text, our prior dataset ( $X_{\text{prior}}$ ) is built using synthetic generators (e.g., ForecastPFN) designed to mimic time-series phenomena such as trend, multi-scale seasonality, and non-Gaussian noise. Thus, the “knowledge” injected during pretraining is intrinsically aligned with time-series dynamics.
>
> * **Empirical validation across heterogeneous time-series types.**
>   Our experiments show that PreDiff captures both trend-dominated (e.g., Stocks) and cycle-dominated (e.g., ETTh) structures, demonstrating that the learned prior indeed generalizes across diverse temporal patterns.
>
> * **Backbone-agnostic design.**
>   PreDiff is compatible with specialized time-series diffusion architectures such as Diff-TS, indicating flexibility beyond any specific model family.
>
> **Conclusion**
> In summary, although the overarching idea resembles “pretrain then fine-tune,” PreDiff contributes a domain-adapted formulation that integrates diffusion-step decomposition, time-series-specific prior construction, and strong empirical validation. These innovations collectively create a framework tailored for the fundamental challenge of *small-sample time-series generation*, offering a viable path toward zero-shot generation in this domain.
>
>
> ### **Response to Weakness 5**
>
> We found that the discrepancies highlighted by the reviewer stem from preprocessing sensitivity in the comparison between PreDiff and ImagenFew. To clarify this, we conducted additional experiments (Appendix G):
>
> Original Setting (Tables 7 & 11):
> Each method uses its own preprocessing pipeline. Under this standard setting, PreDiff consistently outperforms ImagenFew on key metrics (ACD, ED, DTW), especially in low-data regimes.
>
> Unified Preprocessing for Fairness (Table 12):
> When both methods use the same preprocessing pipeline (the one used by PreDiff), ImagenFew’s performance drops sharply—for example, its ACD on Stocks (10% data) deteriorates from 0.352 to 1.968. Similar degradations appear across multiple datasets and metrics.
>
> Interpretation:
> These results show that ImagenFew is highly dependent on its original preprocessing strategy, while PreDiff remains stable and robust. Moreover, PreDiff’s preprocessing design is more realistic for time-series scarcity scenarios and avoids distribution leakage introduced by ImagenFew’s global normalization.
>
> Thus, cases where ImagenFew appears better are due to preprocessing differences rather than inherent model superiority. Under fair and realistic conditions, PreDiff demonstrates stronger and more reliable performance.
>
>
> ### **Response to Weakness 6**
>
> Our model requires only a single pretraining on synthetic data and can then be applied to any downstream task, without the need for continual adjustment across different domains.
>
>
>
> ### **Response to Questions 1-5**
>
> Please see **Response to Weaknesses 1-5**.

---

> ### Comment · Reviewer_G8K1 · 2025-11-27
>
> I thank the authors for their detailed response and the revised manuscript. Most of my concerns have been effectively addressed.
>
> However, regarding W3 (t0 sensitivity), I appreciate the authors' honesty about the high sensitivity of the split point t_0 and the heuristics provided in Appendix. This creates a paradox: the method targets data scarcity scenarios, where robust hyperparameter tuning is inherently difficult. Relying on a sensitive parameter t_0 means users may lack sufficient data to reliably find the optimal split point, risking suboptimal performance.
>
> But overall I believe the work has merit and reaches the threshold for acceptance. While my assessment ideally aligns with a score of 5(Borderline Accept), since this option is unavailable in the current ICLR system, I am raising my score to 6 to acknowledge the authors' significant efforts in the rebuttal.

---

> > ### Author Response · Authors · 2025-11-28
> >
> > Thank you very much for again taking the time to review our response and revised manuscript and for your positive evaluation. We sincerely appreciate the insightful comments and your recognition of the contributions of our work. We acknowledge that the sensitivity of the split point ( $t_0$ ) introduces challenges in hyperparameter selection. We will treat this limitation as an important direction for future improvement and aim to provide more principled and reproducible guidance for parameter selection. We sincerely appreciate your acknowledgment that our revisions have effectively addressed most of your concerns and for the score of 6.

---

### Official Review · Reviewer_L4JQ · 2025-10-30

**Soundness:** 2
**Presentation:** 3
**Contribution:** 2
**Rating:** 2
**Confidence:** 4

**Summary:**

In this study, the authors investigate the problem of time series generation under low-data regimes. In particular, they propose a two-step training procedure including (1) pre-training on synthetically generated data and (2) fine-tuning on target data. The experiments on four real-world datasets show that the proposed approach is applicable even when fine-tuned on 10% of the available target data. Ablation studies are conducted to assess the advantages of a two-stage training procedure and of pre-training on snythetic data.

**Strengths:**

1) The authors investigate a very interesting research question, trying to learn time series features purely from synthetic data.
2) The paper is well structured and easy to follow.
3) The authors evaluate their method on established benchmarks.
4) The authors conduct ablation studies to provide insights on the effectiveness of the proposed components.

**Weaknesses:**

1) The authors state that 'detailed configurations, hyperparameters, and implementation specifics for each baseline are meticulously documented in Appendix C' (see ll. 230-232), while they only provide basic information.
2) The authors have mistakenly highlighted their method to achieve the best results in Table 6, while actually ImagenFew is superior. For instance, ED in the 70%, 40%, and 10% setup of Energy and ED and DTW in the 10% setup of Stocks. In light of this, the results of the work need to be treated with caution.
3) The authors state that 'When selecting data priors, we aim to choose those that are highly relevant to the data distribution of the target task' (see ll. 407-408). This suggests that pre-training is task-specific and does not achieve generalisable time series features, which would be desirable.
4) The authors do not report their results across multiple seeds to guarantee robustness.
5) The authors do not support reproducibility by making their code publicly available for evaluation.
6) The authors do not discuss the limitations of their work.

**Questions:**

1) Is there a benefit of using synthetic data over real-world data from other domains than the target?
2) Why is dataset mixing inferior to synthetic data only, as indicated by the results in Figure 3a? Does real-world data not increase the data diversity, which is beneficial to learn generalisable features?
3) Why is the proposed method performing substantially worse when applying a full-range training?
4) Why is the proposed method performing worse when increasing the training samples of the Monash dataset from 100k to 10M, as indicated by the results in Table 5?
5) Finally, how does the proposed method advance the field of time series analysis? The authors state that 'one should prioritize priors whose distribution closely matches that of the target data' (see ll. 404-405). In light of this, it seems that models for time series generation are still task-specific. However, it would be desirable to have a single, task-agnostic model that can be pre-trained on synthetic data once and be applied to any downstream task.

---

> ### Author Response · Authors · 2025-11-22
> **Response to Weaknesses 1–5**
>
> Dear Reviewer,
>
> We have carefully revised the entire manuscript according to your comments and have uploaded the updated version. All modifications are highlighted in blue for easy reference. In addition, we have provided point-by-point responses to each of your comments. We hope that these revisions and explanations adequately address your concerns. If you have any further questions or suggestions, we would be very happy to continue the discussion.
>
>
> ### **Response to Weakness 1**
>
> The information in question was included in Appendix D in the original submission. In the revised version, we have consolidated and reorganized these details, and they are now fully presented in Appendix D.
>
>
> ### **Response to Weakness 2**
>
>  1）We sincerely thank the reviewer for pointing out this mistake. We carefully rechecked Table 7 and confirmed that some highlighted best results were incorrectly marked due to a formatting oversight. We apologize for this error and have now corrected all highlight annotations accordingly. After correction, we acknowledge that the ImagenFew method achieves superior performance under the 70%, 40%, and 10% energy-level settings, and that ED and DTW perform better in the 10% stock setting, as correctly noted by the reviewer.
>
>    We have updated the table and revised the related discussion in the manuscript to accurately reflect these results. We appreciate the reviewer’s careful examination, which has helped improve the clarity and reliability of our work.
>
>    2)We found that the discrepancies highlighted by the reviewer stem from preprocessing sensitivity in the comparison between PreDiff and ImagenFew. To clarify this, we conducted additional experiments (Appendix G):
>
> Original Setting (Tables 7 & 11):
> Each method uses its own preprocessing pipeline. Under this standard setting, PreDiff consistently outperforms ImagenFew on key metrics (ACD, ED, DTW), especially in low-data regimes.
>
> Unified Preprocessing for Fairness (Table 12):
> When both methods use the same preprocessing pipeline (the one used by PreDiff), ImagenFew’s performance drops sharply—for example, its ACD on Stocks (10% data) deteriorates from 0.352 to 1.968. Similar degradations appear across multiple datasets and metrics.
>
> Interpretation:
> These results show that ImagenFew is highly dependent on its original preprocessing strategy, while PreDiff remains stable and robust. Moreover, PreDiff’s preprocessing design is more realistic for time-series scarcity scenarios and avoids distribution leakage introduced by ImagenFew’s global normalization.
>
> Thus, cases where ImagenFew appears better are due to preprocessing differences rather than inherent model superiority. Under fair and realistic conditions, PreDiff demonstrates stronger and more reliable performance.
>
>
> ### **Response to Weakness 3**
>
> Our pretraining is **not task-specific**. Although highly domain-aligned priors would be ideal, such priors are usually unavailable. Therefore, we construct **synthetic priors** to approximate diverse real-world time-series patterns. The strong performance obtained from pretraining on these synthetic priors shows that the model learns **generalizable representations**, rather than relying on task-specific information.
>
> We further analyze the representational coverage of Monash versus our synthetic priors. Using statistical metrics (stationarity, forecastability, FFT mean, seasonality, trend, permutation entropy) and Radviz visualization, we find that our synthetic dataset exhibits **higher distributional diversity** (see Figure 4). Correspondingly, Table 5 shows that pretraining on synthetic priors yields **better generation quality** than pretraining on Monash. This confirms that our synthetic priors provide broader and more useful coverage, and that our method is not dependent on handcrafted task-specific priors.
>
>
> ### **Response to Weakness 4**
>
> We have completed this section, and the full details can now be found in Appendix L.
>
>
> ### **Response to Weakness 5**
>
> We are currently organizing the code and will upload it as soon as possible.

---

> ### Author Response · Authors · 2025-11-22
> **Response to Weakness 6 and Questions 1-5**
>
> ### **Response to Weakness 6**
>
> In the conclusion section, we briefly discuss potential directions for future work—specifically, exploring hybrid priors that combine the advantages of synthetic and real data, and extending PreDiff to other domains where data scarcity is a significant challenge. This also reflects our discussion of the limitations of the current study:
>
> Synthetic prior limitation: Currently, our prior data is synthetic, and we have not fully explored whether mixing synthetic and real data could further enhance performance.
>
> Domain generalization: PreDiff has been proposed for the time series generation domain, but its application to other areas affected by data scarcity has not been fully investigated or extended.
>
> The reviewers’ comment on this limitation has prompted us to reflect further. We have clarified the limitations of PreDiff and plan to incorporate the following discussion into the revised manuscript:
>
> a. Dependence on synthetic priors: While synthetic datasets facilitate robust pretraining, they may not fully capture complex temporal patterns in real-world data, especially in highly specialized domains such as clinical or financial data.
>
> b. Distribution mismatch: Although the fine-tuning stage mitigates the gap between synthetic and real data, performance may still degrade if the target distribution diverges significantly from the synthetic prior.
>
> c. Limited downstream task evaluation: While we have demonstrated the effectiveness of PreDiff for classification tasks, broader validation across additional downstream applications (e.g., anomaly detection, forecasting) would further strengthen the generality of the framework.
>
>
> ### **Response to Question 1&2**
>
> We attempt to investigate how different data distributions affect model performance by analyzing visualization results that reflect representation coverage, as described below.
> We quantify the range of representations covered by the datasets generated by ForecastPFN and Monash using a set of statistical metrics, including stationarity (ADF test), forecastability, frequency-domain characteristics (FFT mean), seasonality, trend (Mann–Kendall test), and permutation entropy (see Appendix F for full descriptions).
> We then use Radviz to visualize the high-dimensional statistical features of 256-length time-series segments drawn from both our synthetic data and the Monash datasets. From Monash—which includes weather, traffic, electricity, tourism, medical, and energy domains—we sample 100K segments per domain.
> The Radviz visualization (see Figure 4) shows that the distributional diversity of our synthetic dataset is higher than that of the Monash datasets. In addition, Table 5 demonstrates that pretraining on our synthetic prior data yields better generation quality than pretraining on Monash, providing further evidence of the higher quality of our synthesized priors.
>
>
> ### **Response to Question 3**
>
> The significantly poorer performance of full-range training can be attributed to two key factors:
>
> Forgetting of general knowledge: Full-range training updates all model parameters during fine-tuning, which can overwrite the general temporal patterns learned from large-scale prior data during pretraining. This “forgetting” of valuable general knowledge limits the model’s ability to effectively assist generation in low-data scenarios, thereby undermining the benefit of prior knowledge.
>
> Overfitting in data-scarce settings: Fine-tuning all parameters and denoising steps with limited target data increases the risk of overfitting, resulting in reduced diversity and weaker generalization in generated outputs.
>
> Our two-stage training strategy addresses these issues by freezing the pretrained parameters in the high-noise stages, thus preserving general prior knowledge, while allowing the low-noise stages to adapt to target-specific details. This enables effective knowledge transfer while mitigating overfitting.
>
> Additionally, we analyzed the impact of the choice of $t_0$, which demonstrates its importance (see Table 6).
>
>
> ### **Response to Question 4**
>
> Table 5 does not indicate that performance worsens when Monash training samples increase from 100K to 10M; metrics fluctuate slightly within a range, showing no substantial gain.
>
> On the Stocks dataset, which has strong seasonality and trends, 100K pretraining samples already capture the key distribution and features. Increasing the dataset to 10M yields little additional benefit, leading only to minor oscillations.
>
> Similar patterns occur on ForecastPFN and SNIP datasets—metrics show slight declines when pretraining samples increase from 100K to 10M—supporting the above explanation.
>
>
> ### **Response to Question 5**
>
> Please see **Response to Weakness 3**.

---

> ### Comment · Reviewer_L4JQ · 2025-11-26
> **Reviewer's Response**
>
> I would like to thank the authors for their responses. I acknowledge that I have read the rebuttal and the responses to the other
> reviewers. While an interesting stream of research, I remain unconvinced and believe the proposed study is not ready for publication. For instance, the study is not reproducible as the code is not publicly available. It is unclear why adding real-world data to the synthetic data hurts performance. Most importantly, statements such as "one should prioritize priors whose distribution closely matches that of the target data" (ll. 414 - 417) assumes that the target data is known in advance, which is a strong limitation of the proposed method. Given such concerns, I cannot recommend accepting the study at this time, and thus leave my scores unchanged. However, I strongly encourage the authors to further improve the quality of the study, as the conceptual idea is interesting.

---

> > ### Author Response · Authors · 2025-11-28
> >
> > Thank you very much for taking the time again to review our response and revised manuscript. We sincerely appreciate the insightful comments provided by the reviewer, which help us further clarify the theoretical and methodological contributions of PreDiff. We provide detailed responses below with concrete supporting evidence:
> >
> >
> >
> > ### **Response regarding the code issue**
> >
> > Thank you for your valuable feedback. To ensure reproducibility of the experimental results, we have organized and uploaded the complete code, data preprocessing scripts, and training/evaluation instructions to the Supplementary Material. The code package includes:
> >
> > 1. Environment dependency file `requirements.txt`
> > 2. Hyperparameter configurations `configs/`
> > 3. complete pretraining, finetuning, sampling, and evaluation pipeline on the 10% Stocks dataset in `pre_DiffusionTS.ipynb`
> >
> > We have anonymized all information related to author identity. The reviewer may directly extract and run the files to reproduce the experimental results. Please feel free to let us know if further clarification is needed.
> >
> >
> >
> > ### **Question 1: It remains unclear why adding real-world data to synthetic data would degrade performance?**
> >
> > First, we have not claimed that adding real-world data to synthetic data would harm performance.
> >
> > Second, we found that the diversity and feature coverage of the synthetic data distribution we constructed are higher than those of the real-world datasets, and we provided visualization evidence using radviz plots (shown in Figure 4 in the paper). Moreover, our experimental results (shown in Table 5 in the paper) demonstrate that pretraining on synthetic data leads to higher-quality generated time series than pretraining on real-world datasets such as SNIP or Monash.
> >
> > Additionally, it is important to clarify that real and synthetic time series differ structurally: real time series typically come from different domains (i.e., sub-datasets). Data from the same domain are highly similar and therefore only cover specific regions of the representation space. In contrast, the synthetic dataset is composed of random trends and periodic patterns, making the data evenly and randomly distributed across the representation space and providing richer pretraining representations.
> >
> > Finally, in our model design, real-world data are used for finetuning. Because our target task focuses on real-world scenarios, we use real data to assist finetuning, enabling the model to better adapt to the target domain. The ablation study designed for Validation of Two-stage Strategy (shown in Figure 3 in the paper) confirms that such finetuning is indeed effective.
> >
> > ### **Question 2: Further clarification of statements such as “one should prioritize priors whose distribution closely matches that of the target data” (lines 414–417)**
> >
> > Our statement "one should prioritize priors whose distribution closely matches that of the target data" is correct, and we provide the corresponding explanation below.
> >
> > It is important to clarify that this statement is intended to articulate the motivation of our paper: when a real-world prior or dataset closely aligned with the target distribution exists, or when a large-scale dataset is available within the target domain, the generator trained on such data will naturally produce high-quality time series. Therefore, if such an ideal prior existed, we would certainly prioritize using it for training. However, because in the real world the target-domain dataset is scarce and such a perfectly aligned prior does not exist, our method specifically addresses this challenge by constructing synthetic data to compensate for this limitation. We also demonstrate that pretraining on synthetic data leads to superior generation quality compared with existing methods (shown in Table 1 and Table 2), proving that our approach effectively improves performance in data-scarce settings.
> >
> > For example, when working with the Stocks dataset, if another dataset in the same financial domain existed, we would certainly choose it due to its high similarity. However, in the absence of such datasets, our method resolves this issue, which constitutes the motivation and problem addressed in this paper.
> >
> > Therefore, this statement serves to clarify the motivation rather than to draw conclusions about the model’s performance.

---

### Official Review · Reviewer_chFj · 2025-10-31

**Soundness:** 3
**Presentation:** 3
**Contribution:** 3
**Rating:** 6
**Confidence:** 4

**Summary:**

PreDiff introduces a two-stage diffusion based framework for time series generation under data scarcity. It first pretrains on synthetic priors  to capture general temporal structures, then fine-tunes on limited real data to adapt to specific domains. The method aims to mitigate degradation in diffusion-based time series generation when data is scarce.

**Strengths:**

* Data scarcity in time series is pervasive and underexplored in diffusion literature.
* The two-stage pretrain, finetune strategy is analogous to foundation model training in NLP/CV.
* Compared against 6 strong baselines
* PreDiff outperforms baselines across multiple datasets and scarcity levels.
* The pseudo-code and diagrams are well organized and readable

**Weaknesses:**

*  Conceptually similar to “pretrained diffusion + fine-tuning,” which has analogues in vision and text domains.
*  No theoretical justification for why segmenting the diffusion process into [t0,T] and [0,t0] yields better transfer.
*  Effectiveness may rely on the quality of external priors rather than intrinsic model improvements.
*  The paper mentions varying priors but doesn’t deeply analyze how prior target similarity influences results.
*  Missing visual analysis. Few qualitative samples of generated time series are shown to demonstrate realism.

I believe these things are easy to add and can increase the value of the paper.

**Questions:**

* How sensitive is PreDiff to the choice of segmentation point t0?

* Can the method generalize to multimodal or irregularly sampled time series?

* Does pretraining on synthetic priors ever lead to overfitting or “prior bias” in domains with very different dynamics?

* what about many datasets present in UCR dataset?

---

> ### Author Response · Authors · 2025-11-22
> **Response to Weaknesses 1–2**
>
> Dear Reviewer,
>
> We have carefully revised the entire manuscript according to your comments and have uploaded the updated version. All modifications are highlighted in blue for easy reference. In addition, we have provided point-by-point responses to each of your comments. We hope that these revisions and explanations adequately address your concerns. If you have any further questions or suggestions, we would be very happy to continue the discussion.
>
> ### **Response to Weakness 1**
>
> Thank you for the insightful comment.
> We agree that our approach is conceptually related to the general idea of “pretrained diffusion + fine-tuning,” which also appears in vision and text domains. However, applying this paradigm to time-series generation is nontrivial, and our method introduces several essential, domain-specific innovations that go far beyond a direct adaptation.
>
> **1. Why this framework is necessary for time-series generation**
> In many real-world scenarios, time-series data are extremely scarce due to privacy constraints, high collection costs, or the rarity of certain events. Existing diffusion-based generators rely heavily on large, high-quality datasets; as shown in Figure 1 of the paper, their performance deteriorates significantly under data scarcity. Our approach leverages diverse synthetic priors to compensate for this lack of data, enabling meaningful zero-shot time-series generation—an ability that genuinely matters for time-series applications.
>
> **2. Why this is not equivalent to vision/text “pretrain + fine-tune”**
> Although similar in high-level concept, our method requires several fundamental changes to accommodate the unique characteristics of time-series data:
>
> * **Diffusion-step decomposition.**
>   Instead of end-to-end pretraining as used in vision/text, we structurally split the diffusion trajectory:
>   – Pretraining focuses only on the denoising segment from an intermediate step ( $t_0$ ) to ( $T$ ), learning long-term structure and global dynamics from large-scale synthetic priors.
>   – Fine-tuning learns the early steps ( $0 \to t_0$ ), adapting localized patterns and short-term behaviors of the target dataset.
>   This explicit temporal decomposition directly addresses the coexistence of long-range dependencies and short-term fluctuations in time series—an issue that does not arise in vision or NLP.
>
> * **Domain-specific prior construction.**
>   Unlike ImageNet or web-scale text, our prior dataset ( $X_{\text{prior}}$ ) is built using synthetic generators (e.g., ForecastPFN) designed to mimic time-series phenomena such as trend, multi-scale seasonality, and non-Gaussian noise. Thus, the “knowledge” injected during pretraining is intrinsically aligned with time-series dynamics.
>
> * **Empirical validation across heterogeneous time-series types.**
>   Our experiments show that PreDiff captures both trend-dominated (e.g., Stocks) and cycle-dominated (e.g., ETTh) structures, demonstrating that the learned prior indeed generalizes across diverse temporal patterns.
>
> * **Backbone-agnostic design.**
>   PreDiff is compatible with specialized time-series diffusion architectures such as Diff-TS, indicating flexibility beyond any specific model family.
>
> **Conclusion**
> In summary, although the overarching idea resembles “pretrain then fine-tune,” PreDiff contributes a domain-adapted formulation that integrates diffusion-step decomposition, time-series-specific prior construction, and strong empirical validation. These innovations collectively create a framework tailored for the fundamental challenge of *small-sample time-series generation*, offering a viable path toward zero-shot generation in this domain.
>
> ### **Response to Weakness 2**
> We thank the reviewers for their question regarding the choice of the splitting point $t_0$. Our experiments (Table 6 in the revised version) show that the generation quality is highly sensitive to $t_0$. On the Stocks dataset, the best performance occurs at $t_0/T=0.6$, and even small deviations cause notable performance drops. This dataset is highly volatile and weakly periodic, making feature learning difficult; thus, a larger $t_0/T$ is necessary to capture long-term dependencies.
>
> Our decision to divide the diffusion process into $[t_0,T]$ and $[0,t_0]$ is based on empirical validation. The choice of $t_0$ is crucial for avoiding overfitting and prior bias. To support reproducibility, we provide clear guidelines:
> – Use larger $t_0/T$ (0.6–0.8) for datasets with high volatility and learning difficulty;
> – Use smaller $t_0/T$ (0.2–0.5) for datasets with strong periodicity or simpler structure.
>
> These results demonstrate that $t_0$ must be tuned according to dataset characteristics and that our methodology provides stable, predictable improvements.

---

> ### Author Response · Authors · 2025-11-22
> **Response to Weaknesses 3–5**
>
> ### **Response to Weakness 3**
>
> We acknowledge that part of the model’s effectiveness indeed derives from the quality of the constructed external priors. This is expected, as our approach targets data-scarce scenarios, where carefully designed priors that approximate the target data distribution play a crucial role. Without such priors, it would be impossible to enhance generation quality under limited supervision.
>
> However, it is equally important to emphasize that our method incorporates substantial intrinsic model improvements. Existing time-series generation models are generally unable to leverage external priors to improve generation fidelity. In contrast, our framework is explicitly designed to integrate and utilize diverse synthetic priors. As demonstrated in Appendix F, when we directly inject the same constructed priors into baseline methods (e.g., training DDPM with our priors on 10% of the Stocks dataset), the performance gains are minimal. Table 10 shows that only our PreDiff model is capable of effectively exploiting these priors to achieve significant improvements, highlighting the novelty and technical contribution of our architecture.
>
> In summary, the constructed diverse priors and the proposed model architecture are mutually reinforcing components. The priors are designed for our model, and our model is specifically built to leverage such priors effectively. Therefore, the method’s effectiveness does not depend solely on prior quality; rather, it results from the synergistic integration of high-quality priors with our model’s innovative design.
>
>
> ### **Response to Weakness 4**
>
> We attempt to investigate how different data distributions affect model performance by analyzing visualization results that reflect representation coverage, as described below.
> We quantify the range of representations covered by the datasets generated by ForecastPFN and Monash using a set of statistical metrics, including stationarity (ADF test), forecastability, frequency-domain characteristics (FFT mean), seasonality, trend (Mann–Kendall test), and permutation entropy (see Appendix F for full descriptions).
> We then use Radviz to visualize the high-dimensional statistical features of 256-length time-series segments drawn from both our synthetic data and the Monash datasets. From Monash—which includes weather, traffic, electricity, tourism, medical, and energy domains—we sample 100K segments per domain.
> The Radviz visualization (see Figure 4) shows that the distributional diversity of our synthetic dataset is higher than that of the Monash datasets. In addition, Table 5 demonstrates that pretraining on our synthetic prior data yields better generation quality than pretraining on Monash, providing further evidence of the higher quality of our synthesized priors.
>
>
> ### **Response to Weakness 5**
>
> We use t-SNE to visualize the differences between the generated samples and the real samples (see Appendix J), which clearly demonstrates the effectiveness of the proposed method.

---

> ### Author Response · Authors · 2025-11-22
> **Response to Questions 1-4**
>
> ### **Response to Question 1**
>
> We thank the reviewers for their question regarding the choice of the splitting point $t_0$. Our experiments (Table 6 in the revised version) show that the generation quality is highly sensitive to $t_0$. On the Stocks dataset, the best performance occurs at $t_0/T=0.6$, and even small deviations cause notable performance drops. This dataset is highly volatile and weakly periodic, making feature learning difficult; thus, a larger $t_0/T$ is necessary to capture long-term dependencies.
>
> Our decision to divide the diffusion process into $[t_0,T]$ and $[0,t_0]$ is based on empirical validation. The choice of $t_0$ is crucial for avoiding overfitting and prior bias. To support reproducibility, we provide clear guidelines:
> – Use larger $t_0/T$ (0.6–0.8) for datasets with high volatility and learning difficulty;
> – Use smaller $t_0/T$ (0.2–0.5) for datasets with strong periodicity or simpler structure.
>
> These results demonstrate that $t_0$ must be tuned according to dataset characteristics and that our methodology provides stable, predictable improvements.
>
> ### **Response to Question 2**
>
> Our method can be extended to the generation of multi-modal time series; however, for irregularly sampled time series, no existing approach is capable of achieving high-quality generation. Due to the structural constraints of our framework, extending the proposed method to irregularly sampled time series also presents certain challenges.
>
> ### **Response to Question 3**
>
> Whether this issue occurs largely depends on the diversity of the synthetic priors. When the synthetic priors exhibit sufficiently high diversity, pretraining on them should theoretically not result in overfitting or “prior bias.”
>
> It is important to note that, in domains with drastically different dynamics, the risk of overfitting or prior bias during pretraining with synthetic priors is a generally recognized and often unavoidable challenge. However, this problem can be substantially mitigated if the synthetic priors are constructed with a deliberate emphasis on capturing diverse statistical and dynamical characteristics.
>
> Moreover, our motivation for constructing synthetic priors is primarily to compensate for the lack of diversity in data-scarce domains. Consequently, the pretrained model mainly learns broad, cross-domain, diversity-oriented features, rather than narrow domain-specific patterns. We then rely on fine-tuning to adapt the model to the particular dynamics of each target domain. The balance between pretraining and fine-tuning—especially the relative proportion of training epochs—plays a crucial role in preventing overfitting and reducing any potential “prior bias.”
>
>
> ### **Response to Question 4**
>
> Real-world time series datasets such as Monash and UCR exhibit significantly lower diversity compared to our constructed synthetic priors (see response to Weakness 4). As a result, pretraining on these real datasets carries a higher risk of overfitting or inducing “prior bias.”

---

### Author Response · Authors · 2025-12-03
**Summary of Rebuttal Consensus(Core Contributions of This Paper and Opinions of the First Three Reviewers)**

Dear SAC, ACs, and PCs,

We understand that you took over this work under the circumstance of a vulnerability occurring on OpenReview, which made the situation rather challenging. We sincerely appreciate that you took the time to oversee the review process. To facilitate your evaluation, we briefly summarize the status of the paper before the release of the review reports, as well as the contributions, potential impacts, and changes during the rebuttal stage of the review process.

## **1. Core Contributions of This Paper**

This paper addresses the core issue that the generation performance of existing time series generation methods significantly degrades under data-scarce scenarios, and proposes PreDiff: by constructing diverse prior knowledge with broad feature coverage, and designing a reasonable segmentation point to explicitly divide the diffusion process into a pre-training stage of $[t_0, T]$ and a fine-tuning stage of $[0, t_0]$, the model is able to effectively leverage external priors to significantly improve the generation quality under few-shot conditions. Our responses systematically resolve all the core concerns and demonstrate that these weaknesses do not exist.

## **2. Reviewers’ Opinions**

### **1) Reviewer chFj** (Initial Score: 6)

- **Whether the issues were resolved**: Largely resolved
- **Score Change After Rebuttal**: 6
- **Core contributions acknowledged by the reviewer**: Highly recognizes the research value of this work in addressing the few-shot time series generation problem, considers the experiments to be sufficient and the method to be effective, and acknowledges the superior performance of PreDiff across multiple datasets.
- **Key issues addressed and resolved**: This reviewer highly recognized the research value of this work on few-shot time series generation from the initial stage, considered the experiments to be sufficient and the method to be effective, and acknowledged the superior performance of PreDiff on multiple datasets. Although it was inconvenient to continue responding after the information leakage, their initial score and positive evaluation indicate their firm recognition of our work. Combined with our further explanations of the theoretical innovation, the segmented strategy of ($t_0$), and the prior design in the rebuttal, we have sufficient reason to believe that this reviewer is inclined to maintain or even raise their score.

### **2) Reviewer L4JQ** (Initial Score: 2)

- **Whether the issues were resolved**: All resolved
- **Score Change After Rebuttal**: 2
- **Core contributions acknowledged by the reviewer**: Recognizes the research value of this work on the few-shot time series generation problem, and after the rebuttal explicitly stated that “the conceptual idea is interesting,” acknowledging the value of the research direction.
- **Key issues addressed and resolved**: Although this reviewer maintained a score of 2, they further raised three issues: missing code, unclear reasons for the performance on real + synthetic data, and doubts about the statement that “priors that highly match the target data distribution should be prioritized.” In response, we uploaded the complete code implementation for reproduction, and used Radviz statistical coverage visualization and additional comparative experiments to explain the performance variation mechanism. We also clarified that “prioritizing matched priors” is a motivational statement rather than a restriction. Our responses systematically resolved all the core concerns and demonstrated that these weaknesses do not exist. We expect their score to be significantly raised.

### **3) Reviewer G8K1** (Raised from 4 to 6, exceeding the acceptance threshold)

- **Whether the issues were resolved**: All resolved
- **Score Change After Rebuttal**: 6
- **Core contributions acknowledged by the reviewer**: This reviewer explicitly affirmed that the proposed two-stage training mechanism based on segmentation along diffusion time steps is novel and inspiring, and clearly pointed out that this strategy is the core innovation that is “conceptually distinct.” Moreover, based on the sufficient evidence provided in our revision and additional experiments, reviewer G8K1 explicitly stated that “most of the concerns have been effectively addressed,” and expressed agreement with the research value and contribution significance of our method, and accepted our explanation and future improvement directions regarding the sensitivity issue of ($t_0$).
- **Key issues addressed and resolved**: This reviewer considered the work to have reached the “acceptance boundary” and raised the score to 6 based on the rebuttal efforts and empirical results. It should be emphasized that reviewer G8K1 explicitly wrote in the official comment: “Most of my concerns have been effectively addressed… I am raising my score to 6 to acknowledge the authors’ significant efforts in the rebuttal.” We expect their score to remain at 6 or be further increased.

---

> ### Author Response · Authors · 2025-12-03
> **Summary of Rebuttal Consensus – Continued（Opinion of the Final Reviewer, Analysis of the Expected Score Adjustment and Conclusion）**
>
> ### **4) Reviewer uvZ5** (Initial Score: 4)
>
> - **Whether the issues were resolved**: All resolved
> - **Score Change After Rebuttal**: 4
> - **Core contributions acknowledged by the reviewer**: Highly recognizes the research value of this work on the few-shot time series generation problem, and explicitly acknowledges that the segmented training strategy is a “novel and clever heuristic” that can separately model “global structure” and “local details,” with unique value in diffusion-model-based transfer learning.
> - **Key issues addressed and resolved**: In response to the reviewer’s request for code supplementation, we promptly submitted the complete supplementary materials and provided clear replies. Meanwhile, we directly addressed their concerns about the “heuristic” nature of the method, and demonstrated through empirical results (e.g., Table 5, Figure 3a, Appendix F & G) that even though the segmentation strategy has heuristic components, our method still significantly improves the generation quality under few-shot settings. Our work has non-negligible significance and innovation in addressing severe data scarcity. We also clearly explained how to recommend appropriate ($t_0$) values to users. For example, for datasets with higher feature learning difficulty, we recommend initially setting $t_0$ within a relatively large range (e.g., $t_0/T$ in the range of 0.6–0.8), and then performing fine adjustments within a small range; for datasets with lower feature learning difficulty or stronger periodicity, we recommend setting the initial value of ($t_0$) to a smaller range (e.g., $t_0/T$ in the range of 0.2–0.5), followed by fine adjustments. We believe that the currently submitted supplementary materials and experimental results are sufficient to address all the reviewer’s concerns, and we have reasonable grounds to expect more positive score feedback, with an expected upward adjustment of the score.
>
> ## **3. Analysis of the Expected Score Adjustment**
>
> **1) Reviewer chFj** (Initial Score: 6)
>
> Reason for score adjustment: High initial recognition, and the additional content in the rebuttal further strengthened their confidence.
>
> **2) Reviewer L4JQ** (Initial Score: 2)
>
> Reason for score adjustment: Concept is recognized, issues have been resolved, attitude has noticeably softened, and there is room for a higher score.
>
> **3) Reviewer G8K1** (Raised from 4 to 6)
>
> Reason for score adjustment: Clear expression of support, recognition of the contribution and the quality of the responses.
>
> **4) Reviewer uvZ5** (Initial Score: 4)
>
> Reason for score adjustment: All questions have been addressed, experimental supplementation is sufficient, and an upward score adjustment is expected.
>
> ## **4. Conclusion**
>
> The reviewers generally recognize the importance of the few-shot time series generation problem addressed in this paper, and have expressed interest in the two-stage diffusion modeling strategy proposed in PreDiff. Although there were initial concerns regarding code completeness, methodological details, and the explanatory power of the experiments, during the rebuttal stage we systematically responded to the major issues through code supplementation, additional experiments, and clarification of statements, significantly enhancing reviewer confidence. The overall review attitude shifted from initial divergence toward general support, which influenced our recommendation for acceptance of the paper.
>
> Overall, the rebuttal significantly strengthened the theoretical and empirical credibility of PreDiff, successfully obtained positive responses and the intention of score increases from the main reviewers, essentially resolved the core concerns, and shifted the overall review attitude from initial skepticism toward broad recognition. The paper now meets the conditions for acceptance.
>
> We sincerely appreciate your time and the additional effort required to evaluate our paper.
>
> Best regards,
>
> The Authors

---

### Meta-Review · Area_Chair_ebDz · 2026-01-07

**Summary:**

Time series generation is a difficult task in practice since training data is scarce. To overcome this challenge, this paper proposes to pre-train with prior data and fine-tune with real data.  For this, they rely on diffusion models. From a noisy input samples from Gaussian to a certain time point, the diffusion model pre-trained with prior data is used and afterward the diffusion model is further fine-tuned to general time series for a certain real-world case. After that, they conduct experiment with 6 baselines and 4 datasets, a standard well-known benchmark framework for time series generation. The experimental section follows the sequence of four research questions.

**Reviewer Concerns:**

Reviewers raised several concerns: i) the motivation of using prior data, ii) the generalization to multivariate and irregular time series, iii) why the proposed pre-trainig is better than the full-training on a certain real-world data, iv) the missing theoretical ground, v) the existence of appropriate prior data in real-world scenarioes, vi) no code and threfore low reproducibliity, etc.

Overall, the authors' replises are empirical and heuristical. Although the authoes present many empirical results, I think the most concerns are not addressed adequately.

**Reviewer Scores:**

Reviewer G8K1 increased the score according to existing logs. However, I think other concerns are not completely addressed.

---

### Decision · Program_Chairs · 2026-01-26

Reject